# Nuclear fate of yeast snoRNA is determined by co-transcriptional Rnt1 cleavage

Pawel Grzechnik[1,2], Sylwia A. Szczepaniak[3,4], Somdutta Dhir[1], Anna Pastucha[3], Hannah Parslow[2], Zaneta Matuszek[3], Hannah E. Mischo[1], Joanna Kufel[3] & Nicholas J. Proudfoot[1]

Small nucleolar RNA (snoRNA) are conserved and essential non-coding RNA that are transcribed by RNA Polymerase II (Pol II). Two snoRNA classes, formerly distinguished by their structure and ribonucleoprotein composition, act as guide RNA to target RNA such as ribosomal RNA, and thereby introduce specific modifications. We have studied the 5′end processing of individually transcribed snoRNA in *S. cerevisiae* to define their role in snoRNA biogenesis and functionality. Here we show that pre-snoRNA processing by the endonuclease Rnt1 occurs co-transcriptionally with removal of the m$^7$G cap facilitating the formation of box C/D snoRNA. Failure of this process causes aberrant 3′end processing and mislocalization of snoRNA to the cytoplasm. Consequently, Rnt1-dependent 5′end processing of box C/D snoRNA is critical for snoRNA-dependent methylation of ribosomal RNA. Our results reveal that the 5′end processing of box C/D snoRNA defines their distinct pathway of maturation.

[1] Sir William Dunn School of Pathology, University of Oxford, South Parks Road, Oxford OX1 3RE, UK. [2] School of Biosciences, University of Birmingham, Edgbaston, Birmingham B15 2TT, UK. [3] Institute of Genetics and Biotechnology, Faculty of Biology, University of Warsaw, Pawinskiego 5a, 02-106 Warsaw, Poland. [4] College of Inter-Faculty Individual Studies in Mathematics and Natural Sciences, University of Warsaw, 02-089 Warsaw, Poland. These authors contributed equally: Pawel Grzechnik, Sylwia A. Szczepaniak. Correspondence and requests for materials should be addressed to P.G. (email: P.L.Grzechnik@bham.ac.uk) or to J.K. (email: kufel@ibb.waw.pl) or to N.J.P. (email: nicholas.proudfoot@path.ox.ac.uk)

Small nucleolar RNA (snoRNA) are classified based on conserved RNA sequences as either box C/D or H/ACA snoRNA[1]. These sequences form specific secondary structures which are associated with class-specific proteins. Small ribonucleoprotein complexes (snoRNP) are essential for many cellular processes including RNA processing, mRNA splicing, chromatin maintenance and RNA editing[2,3]. However, their predominant function is in ribosome biogenesis where they guide ribosomal RNA (rRNA) 2′-O-methylation (box C/D snoRNA) and pseudouridinylation (box H/ACA snoRNA)[2].

Genomic organization of snoRNA genes differs between organisms. In higher eukaryotes they exist as polycistronic transcription units (TUs) or are located within the introns of protein-coding genes. Few are expressed as single independent genes[4–7]. In Saccharomyces cerevisiae (S.cerevisiae) they are predominantly independent TUs with only a minority as polycistrons or within pre-mRNA introns[8]. All yeast snoRNA, except Pol III-dependent SNR52, are Pol II transcribed with their transcription termination mediated by the combined action of Nrd1-Nab3-Sen1 (NNS) and cleavage and polyadenylation complexes[9–12]. This is followed by 3′end processing with rounds of oligoadenylation and exonucleolytic digestion performed by TRAMP4/5, the core nuclear exosome and associated exonuclease Rrp6[13,14].

Like 3′end processing, snoRNA 5′end maturation in yeast is distinct from mRNA processing since mature snoRNA lack the mRNA-specific m$^7$G cap (Fig. 1a). Most box C/D pre-snoRNA are synthesized with a 5′capped extension forming a stem-loop with the AGNN loop consensus sequence. The endonuclease Rnt1 recognizes and cleaves this stem-loop at specific Rnt1 cleavage sites (RCS)[15,16]. The remaining, uncapped 5′extensions are further processed by Rat1 and Xrn1 exonucleases. In some instances 5′end maturation may occur independently of Rnt1 cleavage, by exonucleolytic trimming[16]. All snoRNA clusters are box C/D class and excised from their TU by Rnt1, thereby removing the m$^7$G cap structure from the first snoRNA[15,17–19]. Only four box C/D snoRNA are transcribed without a 5′end extension like most box H/ACA snoRNA. Here the 5′end cap structure remains on the mature snoRNA, but is converted into trimethylated m$^{2,7,7}$G (TMG) cap by Tgs1 to distinguish it from mRNA caps (Fig. 1a)[15,20].

In S. cerevisiae physical interaction between the Cap Binding Complex (CBC) and NNS has linked 5′cap structure with 3′end processing[21]. Similarly, in human cells, CBC collaborates with the Nrd1 orthologue ARS2 in transcriptional termination and 3′end processing of small nuclear RNA (snRNA)[22,23]. This suggests that in both eukaryotes 5′ and 3′end processing may be functionally interdependent for capped small non-coding RNA. We have set out to test if 5′end processing of Rnt1-dependent (RD) snoRNA (mainly box C/D) is also coupled to its 3′end processing. This was an attractive hypothesis as Rnt1 has been also reported to interact with the NNS complex[24]. We show that removal of the mRNA-like 5′end cap structure is required for RD box C/D snoRNP maturation and function. Furthermore, Rnt1 co-transcriptionally cleaves the 5′end of snoRNA precursors removing the 5′capped extension. This facilitates further steps in box C/D snoRNA maturation. Notably, lack of snoRNA 5′end processing affects multiple downstream events, including 3′snoRNA trimming and nuclear localization, and ultimately inhibits rRNA methylation. This connection between maturation and functionality may explain why intron-encoded snoRNA, lacking independent transcription start sites, have evolved in higher eukaryotes.

## Results

### Co-transcriptional 5′end processing of snoRNA precursors.
The mature 5′ends of both box H/ACA and C/D snoRNA are generated either by Rnt1-dependent cleavage followed by exonucleolytic trimming or by m$^7$G cap trimethylation (Fig. 1a)[15,16]. To systematically catalogue snoRNA 5′end processing, we employed RNA-seq analysis to detect 5′extended pre-snoRNA in the rnt1Δ strain. We identified three additional box H/ACA (snR81, 83, 85) and one box C/D snoRNA (snR87) with 5′extensions that form Rnt1 cleavage structures (Supplementary Fig. 1a). Combining this and previously published data[15–19], we show that each snoRNA class generally employs a different 5′end processing pathway (Fig. 1b and Supplementary Table 1). The 5′ends of box C/D snoRNA are generated by Rnt1 dependent (RD) cleavage with only four snoRNA out of 24 independent and five polycistronic genes retaining caps. In contrast, 21 box H/ACA snoRNA have capped 5′ends with only six cleaved by Rnt1 at their 5′ends. These results indicate that cap and associated proteins play different roles in the biogenesis of each snoRNA class.

In human cells, CBC interacts with transcription termination factors and is known to be involved in 3′end processing of small nuclear RNA (snRNA)[22,23]. Similarly, both yeast CBC and Rnt1 interact with the NNS termination complex[21,24]. We therefore established the profiles of CBC and Rnt1 across all snoRNA genes in S. cerevisiae using ChIP-seq analysis. Yeast strains were generated with endogenous myc-tagged RNT1 and CBC20 genes. Since Cbp20 and Cbp80 form a heterodimer[25], the presence of Cbp20 is indicative of the whole CBC complex. Notably, CBC is recruited across the entire TU of all box C/D and H/ACA snoRNA (Fig. 1c,d and Supplementary Fig. 1b), as with protein-coding genes[26]. Instead, Rnt1 is more enriched towards the 3′ends of RD snoRNA TUs (Fig. 1c,d and Supplementary Fig. 1b), downstream of Nrd1 binding sites (NBS)[27]. Differential Cbp20 and Rnt1 co-transcriptional binding profiles are especially evident for longer polycistronic snoRNA TUs (Fig. 1d and Supplementary Fig. 1b). Thus, CBC peaked in the proximal parts of polycistrons while Rnt1 accumulated over transcription termination regions, even though RCS are proximal to transcription start sites (TSS). 3′end-specific Rnt1 recruitment may rely on interaction with snoRNA associated proteins and NNS[24,28,29] since it is also detectible on Rnt1-independent box C/D and H/ACA snoRNA as well as over NNS-dependent protein coding genes (Fig. 1e and Supplementary Fig. 1c,d). Rnt1 was also present over many protein-coding genes (see database), suggesting broader functions for Rnt1[30,31]. In contrast, CBC did not show a recruitment bias towards gene 3′ends and NNS binding sites (Fig. 1c,d and Supplementary Fig. 1b). While both CBC and Rnt1 interact with NNS, RNase treatment reduced their ChIP signals over Rnt1-dependent SNR47 (Fig. 1f). Possibly CBC and Rnt1 interact with NNS containing RNA, which is displaced from Pol II CTD prior to transcriptional termination.

We note that CBC signals decreased over the 3′ends of RD snoRNA genes and especially the much longer polycistronic TUs (Fig. 1d) but not for Rnt1-independent snoRNA, such as 606 nt long box H/ACA SNR30 and 194 nt long box C/D SNR4 (Supplementary Fig. 1e). Since Tgs1 is localised in the nucleolus[20] and cap trimethylation most likely occurs post-transcriptionally, the decrease in Cbp20 signal over RD snoRNA gene 3′ends suggests that Rnt1 cleaves pre-snoRNA co-transcriptionally, thereby removing cap associated CBC. To further investigate Rnt1 co-transcriptional cleavage we analysed available NET-seq data[32]. Although NET-seq was primarily developed to map nascent RNA 3′ends in the Pol II active centre, it also detects RNA 3′ends generated by co-transcriptional cleavage within protein complexes associated with Pol II[32,33]. Notably, NET-seq displayed pronounced signals corresponding to known or predicted Rnt1 cleavage sites for both box C/D and H/ACA snoRNA (Fig. 1g and Supplementary Fig. 1f). NET-seq peaks co-localized with 27 out of 29 RCS present in snoRNA 5′extensions

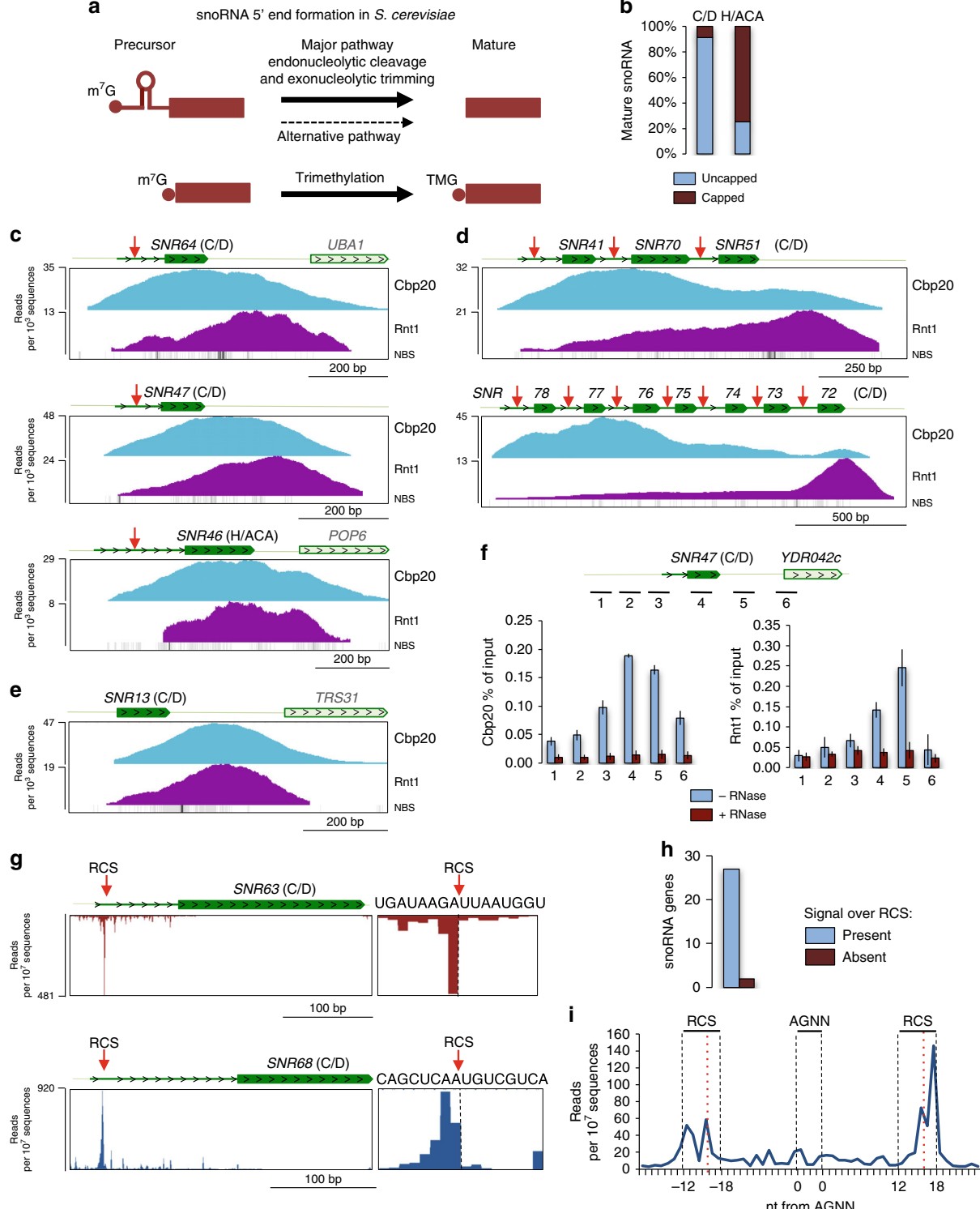

**Fig. 1** Co-transcriptional 5′end processing of snoRNA. **a** 5′processing pathways for independently transcribed snoRNA in *S. cerevisiae*. **b** Mature independently transcribed snoRNA (including snoRNA located at the 5′ends of polycistronic TUs) possessing or lacking caps. **c–e**ChIP-seq analyses of Cbp20 and Rnt1 co-transcriptional recruitment to **c** snoRNA transcribed as 5′extended precursors, **d** polycistronic snoRNA, **e** snoRNA transcribed without 5′extension. NNS-binding sites (NBS) are shown below the reads. **f** ChIP analysis showing RNA-dependent recruitment (RNase sensitive) of Cbp20 and Rnt1 to RD box C/D *SNR47*. Average of three independent experiments is shown. Error bars represent standard deviation. **g** NET-seq signals originating from Rnt1 cleavage sites (RCS) in the pre-snoRNA 5′extensions. Note that NET-seq signals were bioinformatically depleted over sequences coding mature snoRNA. **h** The total number of snoRNA (including proximal snoRNA from each polycistron) displaying NET-seq signal within the RCS in the 5′extension. **i** Metagene analysis of NET-seq peaks around the AGNN loop in the snoRNA 5′extension; RCS –Rnt1 cleavage site, the predicted cleavage site is denoted by red vertical dotted line. Green rectangle denotes mature snoRNA while bold green line shows 5′extension. RCS is marked by red vertical arrow

(24 independent and 5 polycistronic snoRNA genes) (Fig. 1h and Supplementary Table 2). Meta-analysis of snoRNA 5′ends placed these RCS within 12–18 nucleotides of AGNN hairpin loops (Fig. 1i). No NET-seq peaks were detected upstream of snoRNA transcribed without 5′end extensions (Supplementary Fig. 1g). We predict that Rnt1 recruited to RD snoRNA 3′ends remove capped 5′ends from pre-snoRNA containing specific RCS. This implies cross-talk between both ends of the transcription unit.

**Influence of cap and associated CBC on snoRNA expression levels.** To investigate the requirement of cap and associated CBC for snoRNA maturation, a temperature-sensitive *ceg1-63* mutant was employed to inactivate the essential guanyl transferase, Ceg1[34,35]. Promoters of endogenous snR13 (C/D) and snR3 (H/ACA), were replaced by the inducible *GAL1* promoter in WT and *ceg1-63* strains. Note that both snoRNA lack RCS, to avoid unprotected 5′end degradation. Galactose induction of box C/D

*SNR13* in WT at 37 °C accumulated mature (M) and a decapped, 5′truncated (Mt)[36] snR13 as well as previously described 3′extended processing intermediates[14]: shorter oligoadenylated substrates (Me) for the Rrp6 exonuclease and longer poly-adenylated precursors (Pa) processed by both Rrp6 and the exosome (Fig. 2a)[14]. The basal activity of the *GAL1* promoter yielded Mt under all conditions. Its function is unknown[36]. *GAL1* induced *SNR13* in *ceg1-63* cells at non-permissive temperature (37 °C) revealed a similar accumulation of mature snR13 transcripts over time (Fig. 2a,d), but reduced levels of 3′extended pre-snR13 precursors (Me)[14]. This indicates an altered Rrp6-dependent 3′end processing phenotype in this strain. Notably, *RRP6* deletion restored the accumulation of Me snR13 precursors in *ceg1-63* (Supplementary Fig. 2a). Also, precursors were shortened in *rrp6Δ ceg1-63*, suggesting that uncapped precursors are more sensitive to 3′–5′exonucleolytic digestion.

In contrast to snR13, *GAL1* induced box H/ACA snR3 synthesis was severely restricted by *CEG1* mutation at

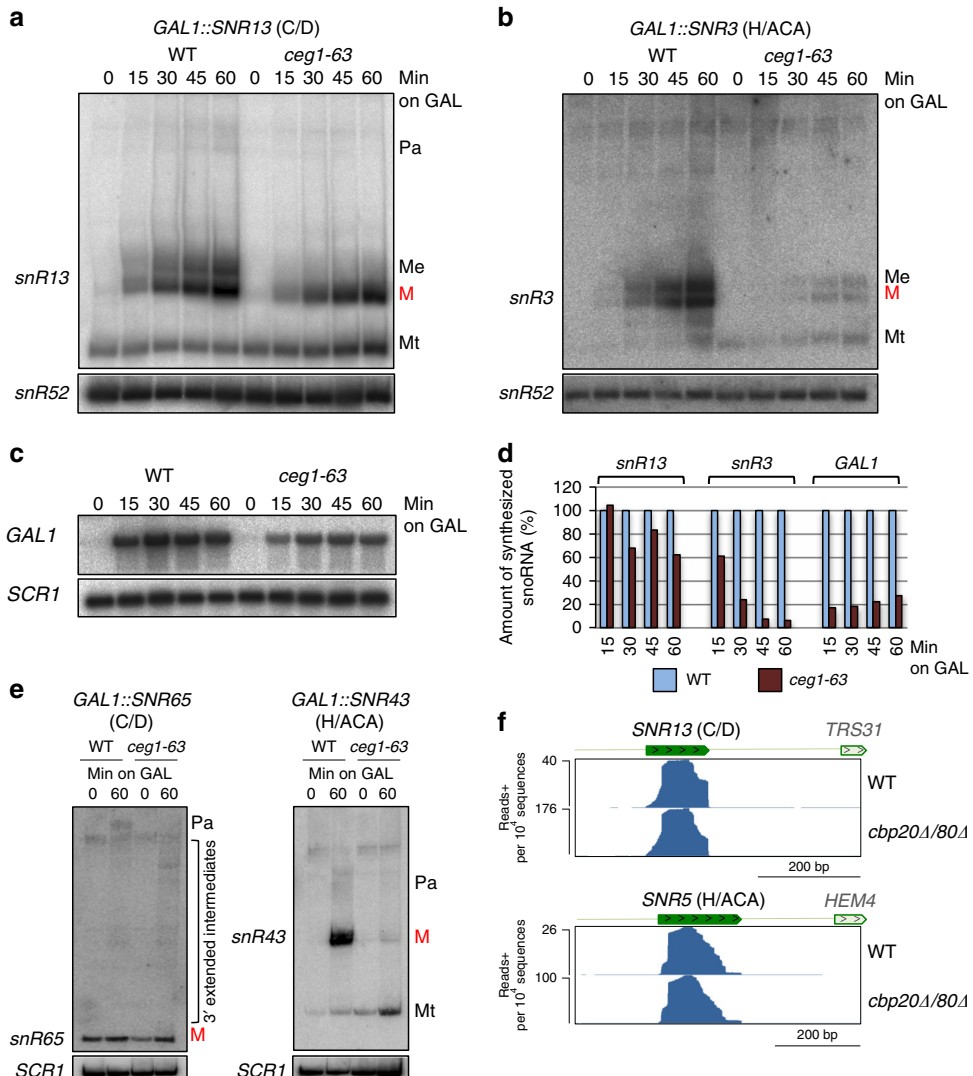

**Fig. 2** Box H/ACA but not C/D snoRNA require cap structure. Transcriptional induction of **a** box C/D *SNR13*, **b** box H/ACA *SNR3* **c** and protein-coding *GAL1* in *ceg1-63* versus WT strains; Northern blot analysis. Cells were shifted to the non-permissive temperature for 15 min prior to galactose induction and were incubated at this temperature during transcriptional pulse. Pa precursors with long polyadenylated 3′extensions, Me precursors with short oligoadenylated 3′extensions, M mature, Mt 5′truncated mature snoRNA. **d** Quantification of synthesis rates for snR13, snR3 and *GAL1* mRNA in WT and *ceg1-63* strains normalised to the loading controls. Values for WT were set to 100%. **e** Transcriptional induction profiles of RD box C/D *SNR65* and box H/ACA *SNR43*; mature and precursors snoRNA are marked as above. Northern blot analysis. **f** RNA-seq analysis of total RNA from *cbp20Δ cbp80Δ* strain. The X axis values are independently adjusted for each sample. Green and light green rectangles denote mature snoRNA and adjacent protein-coding genes, respectively

37 °C (Fig. 2b,d). Compared to WT, total snR3 levels expressed from the GAL1 promoter were substantially lower in ceg1-63. SnR3 precursors were mapped by RNase H treatment followed by Northern blot analysis, confirming that the Me species does not originate from an alternative TSS (Supplementary Fig. 2b). Thus, box H/ACA snR3 behaves like a mRNA, such as GAL1, requiring a cap for transcript stability (Fig. 2c,d)[35]. We also confirmed that at permissive temperature in the ceg1-63 mutant, galactose-induced snR13 was unaffected, while snR3 levels were slightly decreased over 1 h (Supplementary Fig. 2c).

Since SNR13 and SNR3 lack RCS, we also tested RD box C/D SNR65 and SNR68 as well as RD box H/ACA SNR43 and SNR46 transcribed from GAL1 promoters in WT and ceg1-63 cells. GAL1 promoter insertion excluded RCS to prevent possible co-transcriptional degradation from the unprotected 5′end. The accumulation (after 60 min induction) of mature snR65 was low but comparable between WT and ceg1-63 cells (1.5× and 1.78× fold respectively) (Fig. 2e), Moreover, long pre-snR65 were absent and processing intermediates were detected in ceg1-63. Transcription of SNR68 from GAL1 gave very weak accumulation of long 3′extended pre-snR68 (Supplementary Fig. 2d), unsuitable for further analysis. In contrast, GAL1 promoted box H/ACA SNR43 and SNR46 levels were strongly activated by galactose but this was inhibited by CEG1 mutation. Accumulation of all snR43 and snR46 species in ceg1-63 was respectively 5.5× and 4.8× fold lower than in WT (Fig. 2e and Supplementary Fig. 2d). These results indicate that 5′extensions and m[7]G cap play differential roles in RD box C/D and H/ACA snoRNA biosynthesis.

We next tested if yeast CBC is required for snoRNA 3′end processing. RNA-seq analysis detected no unprocessed box C/D or H/ACA snoRNA in the cbp20Δ cbp80Δ mutant (Fig. 2f). Increased snoRNA levels observed in the double mutant (Supplementary Fig. 2e) indicate that CBC influences snoRNA abundance by the regulation of exonucleolytic nuclear degradation[37]. Consistently, ChIP analysis employing anti-Pol II antibody showed unchanged levels of Pol II between WT and cbp80Δ cells (Supplementary Fig. 2f).

Overall, we conclude that the synthesis of box H/ACA snoRNA requires m[7]G cap. In contrast, this structure appears dispensable for the expression of box C/D snoRNA, but may still affect some steps in their 3′end maturation. The reduction of snR13 and snR65 3′extended precursors in ceg1-63 suggests that cap removal from box C/D pre-snoRNA accelerates 3′processing.

**Rnt1 cleavage controls 3′ends of RD box C/D snoRNA.** Our data above suggest that RD box C/D snoRNA have their 5′caps removed co-transcriptionally by Rnt1. Furthermore, cap presence is not required for the expression of box C/D snoR13 and snR65 though it may affect 3′end maturation. We tested by RNA-seq analysis if 5′end processing regulates 3′processing differently between WT and rnt1Δ strain where pre-snoRNA 5′ends remain uncleaved. Notably, boxC/D snoRNA transcribed with 5′extensions as well as the last snoRNA from polycistronic TUs have short unprocessed extensions at their 3′ends in rnt1Δ cells (Fig. 3a, Supplementary Fig. 3a and Supplementary Table 3). In contrast, 3′ends of box C/D snoRNA transcribed without 5′extension and all box H/ACA snoRNA (including snoRNA transcribed with 5′extensions) were unaffected by RNT1 deletion (Fig. 3b and Supplementary Fig. 3a). This RNA-seq analysis was confirmed by Northern blot analysis (Supplementary Fig. 3b). To compare 3′ends in WT and rnt1Δ, we generated homogenous 5′ends by prior RNase H digestion with oligonucleotides complementary to mature snoRNA 5′ends. These data also show that in the rnt1Δ strain, although 5′extended pre-snoRNA are dominant, some 5′processed snoRNA are still

generated (Supplementary Fig. 3b) by an alternative pathway (Fig. 1a).

The proportion of transcripts unprocessed at their 3′end in rnt1Δ was measured by circular RNA RT-PCR (CR-RT-PCR) technique for box C/D snR68 and snR65 to simultaneously sequence the 5′ and 3′ends of the same RNA molecule. This required snoRNA cap removal by RNase H digestion. PCR primers directed against the snoRNA mature sequence allowed detection of both pre- and mature snoRNA (Fig. 3c). CR-RT-PCR on RNA isolated from rnt1Δ showed that most 5′extended box C/D snR68 and snR65 possessed short, oligonucleotide extensions with an oligo(A) tail. In contrast, snoRNA with mature 5′ends (presumably processed by the Rnt1-independent alternative pathway) have normally processed 3′ends. This analysis reveals that removal of RD snoRNA 5′extensions is critical for their 3′end trimming. Also, these RNA are oligoadenylated, implicating TRAMP4/5-activated Rrp6- and exosome-dependent digestion[38]. However, accumulation of oligo(A) tails in the rnt1Δ strain suggests that pre-snoRNA, which have not been processed at their 5′ends, are stalled in 3′–5′exonucleolytic processing.

We finally performed CR-RT-PCR on RNA from the WT strain using oligonucleotides located over the SNR68 coding sequence and 5′extension to focus the analysis on the 3′ends of unstable 5′extended pre-snoRNA (Fig. 3d). All sequenced clones had 3′extension from 2 to ~100 nucleotides, giving a range of precursors that extend to the transcription termination sites. No pre-snR68 with fully processed 3′ends were detected, indicating that snoRNA 5′end processing precedes 3′end processing. CR-RT-PCR with the same primer set but for the rnt1Δ strain detected only short (up to 10 nt) oligoadenylated 3′extensions (Supplementary Fig. 3c).

Since most box H/ACA snoRNA retain their cap structure, which is later trimethylated by Tgs1[15,20], we also tested if lack of Tgs1 affects 3′end formation of snoRNA. However, no 3′end processing defects were detected in the tgs1Δ mutant (Fig. 3e). TGS1 loss did not enhance box C/D-specific 3′end processing defects as observed in the rnt1Δ strain (Fig. 3f). Overall, a clear interplay between Rnt1-dependent 5′end processing and 3′end exosome-dependent processing for RD box C/D snoRNA is evident.

**Cap retention impairs box C/D snoRNA processing.** We wished to determine whether the 3′end processing defect in the rnt1Δ strain is caused by inactivity or absence of Rnt1. SnR68Δ strain was transformed with plasmids expressing either WT or a snR68 stem-loop mutant that is insensitive to Rnt1 processing[39]. Northern blot analysis of RNase H cleaved snR68 showed that the stem-loop mutation results in the same 3′end trimming defect as observed in the rnt1Δ mutant (Fig. 4a and Supplementary Fig. 3b). snR68 processing in the rnt1-E320K catalytic mutant[40] generated aberrantly processed 3′ends, like the rnt1Δ strain (Supplementary Fig. 4a). These results show that RD box C/D snoRNA 3′end processing requires removal of 5′end extensions. We next investigated if unprocessed stem-loops in the 5′extension somehow interfere with 3′end exonucleolytic processing. We therefore generated a strain with the GAL1 promoter and its 5′UTR integrated upstream of box C/D SNR13 (GAL1U::SNR13 strain) (Fig. 4b). GAL1 transcription formed a hybrid GAL1 UTR-snR13 RNA (snR13e). In a second strain, box H/ACA SNR3 was placed between the GAL1 UTR and SNR13 (GAL1U::SNR3::SNR13). Processing of these two artificially extended snoRNA genes was compared to the unmodified snR13 also expressed from a GAL1 promoter, lacking the UTR sequence (GAL1::SNR13) (Fig. 4b). Northern blot analysis revealed that the presence of GAL1 UTR at the 5′end of snR13e inhibited 3′end trimming, while snR13 synthesized without the 5′extension was

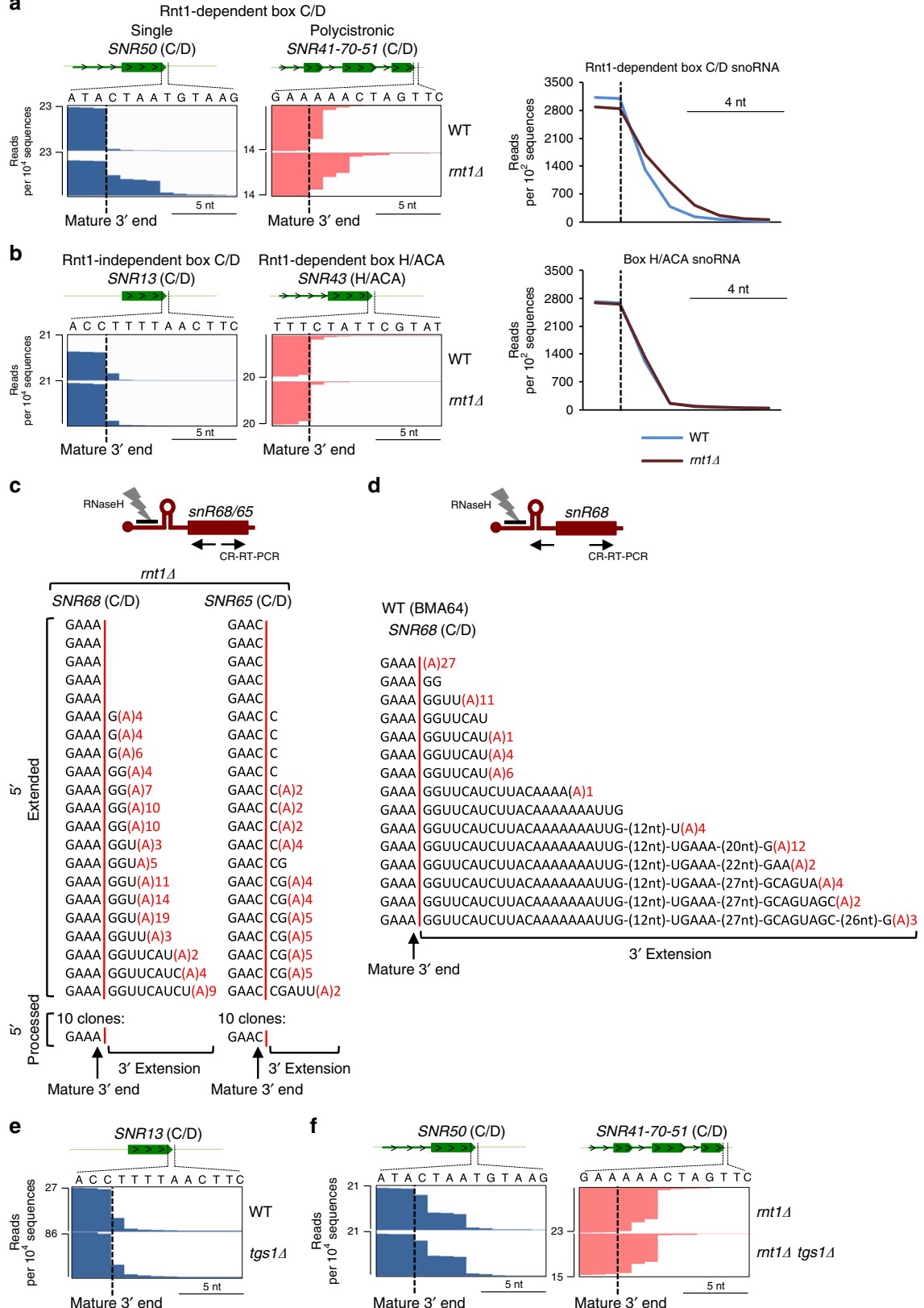

**Fig. 3** Lack of the 5′processing affects box C/D snoRNA 3′end maturation. RNA-seq analysis of **a** RD box C/D snoRNA and **b** Rnt1-independent box C/D snR13 as well as RD box H/ACA snoRNA 3′ends in WT versus *rnt1Δ* strains. Metagene analyses are shown on the right. Dotted vertical lines denote mature 3′end. **c**, **d** CR-RT-PCR analysis as shown in diagram, for snR68 and snR65 in *rnt1Δ* and WT strain. Red vertical lines separate mature 3′ends from 3′extension (post-transcriptionally added oligo(A) in red). **e**, **f** RNA-seq analysis of specified snoRNA 3′ends in *tgs1Δ* and *rnt1Δ/tgs1Δ* strains. SnoRNA used for metagene analysis are listed in Supplementary Table 3

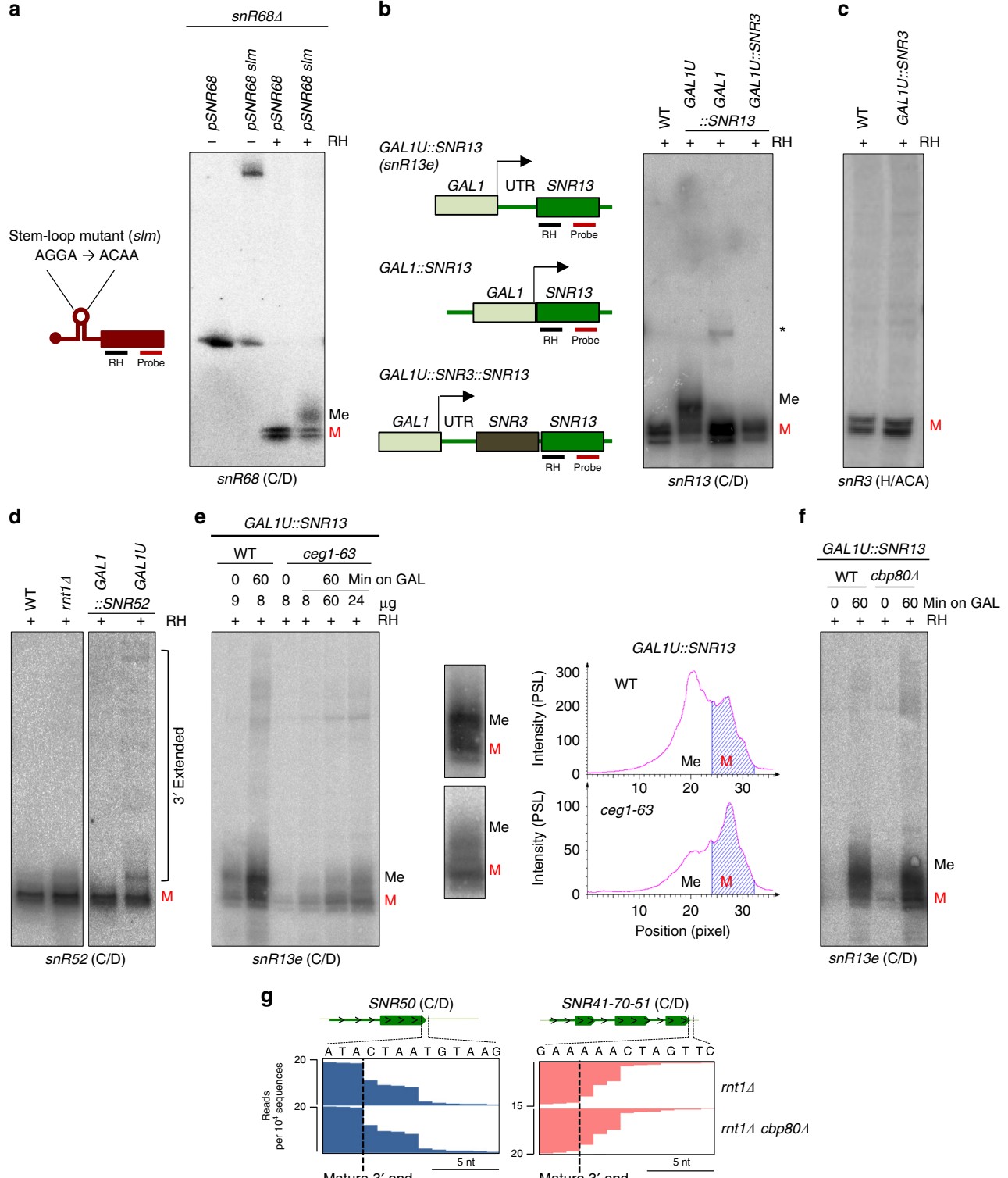

**Fig. 4** Capped 5′extension affects box C/D pre-snoRNA 3′end processing. **a** Northern blot of WT snR68 or mutated in the stem-loop. Oligonucleotides used for RNase H cleavage and detection are indicated on the left. **b** Northern blot of *SNR13* modified strains. Diagram on the left shows variants of endogenous *SNR13* expressed from ectopic *GAL1* promoter and oligonucleotides used for RNase H cleavage and detection. Asterisk marks incompletely digested snR13. **c** Northern blot analysis of *SNR3* expressed from *GAL1* promoter. **d** Northern blot analysis of snR52 transcribed from endogenous (Pol III-dependent) and *GAL1* promoter in *rnt1Δ* and WT strains. **e** Northern blot and quantitation of snR13e transcripts in *ceg1-63* strain. Amount of RNA loaded on the gel is indicated above the radiogram. Blue stripes on the hybridization profiles indicate the area covering 3′processed snR13e. **f** Northern blot of snR13e expressed in *cbp80Δ* strain. **g** RNA-seq showing snoRNA 3′ends in *rnt1Δ cbp80Δ* strain. **a–f** RNA was treated with RNAse H to create homogenous 5′ends. M denotes snoRNA with mature 3′ends; Me denotes snoRNA with 3′end extensions

normally processed. As before, we generated homogenous 5′ends by specific RNase H digestion (Fig. 4b). Surprisingly, 5′extended hybrid box H/ACA snR3- C/D snR13e was also correctly 3′end trimmed. This indicates that the presence of box H/ACA snR3 enables correct processing of *GAL1* UTR-snR3-snR13 dicistronic snoRNA by overriding the adverse impact of a capped 5′extension. Consistently, *SNR3* transcribed with the *GAL1* UTR did not display processing defects compared to WT (Fig. 4c and Supplementary Fig. 4c). It appears that the mere presence of a capped 5′extension affects Rrp6-dependent 3′processing of box C/D, but not H/ACA snoRNA. This is consistent with the *rnt1Δ* strain data where box H/ACA snoRNA 3′end processing is unaffected by the presence of 5′extensions (Fig. 3b).

We next tested if pre-snoRNA cap and CBC disturb the 3′end formation of box C/D snoRNA. Northern blot and RNA-seq analyses show that snR52, transcribed as an uncapped 5′extended precursor by Pol III, is normally processed at the 3′end in the *rnt1Δ* strain (Fig. 4d and Supplementary Fig. 4d,e). Also, snR52 transcribed from the Pol II-dependent *GAL1* promoter

was correctly trimmed at the 3′end when its transcription start was immediately followed by the snoRNA mature sequence. However, artificial extension of snR52 with the *GAL1* UTR as in our modified *snR13* constructs (Fig. 4b) resulted in accumulation of 3′extended RNA (Fig. 4d). Here, 3′unprocessed snR52 were longer and more heterogeneous than with other C/D snoRNA. Possibly Pol III-dependent *SNR52* does not possess a typical NNS terminator, so that 3′extended snR52 species represent read-through transcripts.

To test if lack of m⁷G cap synthesis restores 3′end processing of snR13e, *GAL1U::SNR13* transcription was tested in the *ceg1-63* strain to impair m⁷G cap synthesis[35]. Induced levels of snR13e in *ceg1-63* decreased 3-fold compared to WT (Fig. 4e). Possibly exonucleolytic degradation over the *GAL1* UTR compromised snoRNP formation, resulting in enhanced degradation of snR13e. Read quantitation over the mature and 3′untrimmed snR13e show that in WT, 3′unprocessed snR13e was more abundant than the mature fraction, while in *ceg1-63* this effect was reversed (Fig. 4e). This suggests that a retained cap structure causes the

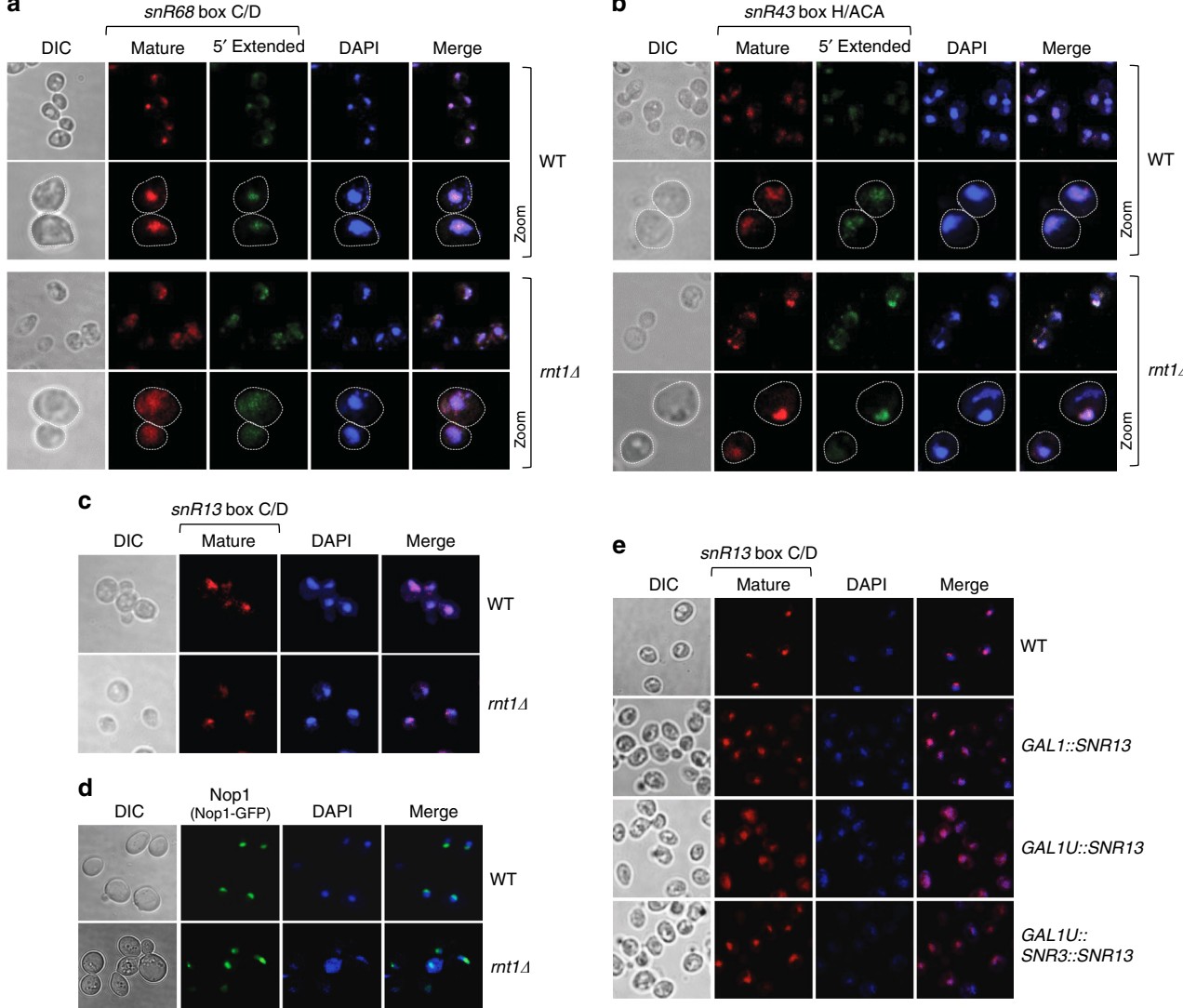

**Fig. 5** 5′extended capped box C/D pre-snoRNA are mislocalized. Localization of **a** Rnt1-dependent box C/D snR68, **b** box H/ACA snR43 and **c** Rnt1-independent box C/D snR13 in *rnt1Δ* versus WT strains; FISH analysis. Probes against snoRNA mature sequence visualize both mature and pre-snoRNA, while probes complementary to the 5′extension detect only 5′extended precursors. **d** Nop1-GFP nuclear localization in the *rnt1Δ* strain. **e** Cellular localization of snR13 expressed from variant extensions of *GAL1* promoter. The same probe against snR13 mature sequence was used for all strains. Nuclei were visualised by DAPI staining. DIC denotes differential interference contrast

snR13e 3′end processing defect. Reduction in the 3′untrimmed species was not observed during snR13e induction in the *cbp80Δ* strain, where CBC functions are compromised (Fig. 4f). Consistently, RNA-seq analysis of *rnt1Δ/cbp80Δ* did not reveal enhanced 3′end processing of RD box C/D snoRNA (Fig. 4g). Overall, we show that m7G cap, but not associated CBC, present on the box C/D snoRNA precursor interferes with the final step of 3′end processing.

**5′end processing controls snoRNA cellular fate**. SnoRNA 3′end processing is mediated by the combined exonucleolytic activities of the exosome core and Rrp6[13,14]. Our results indicate that snoRNA capped 5′extensions block snoRNA 3′end processing. Possibly 5′unprocessed RD box C/D snoRNA are exported to the cytoplasm (mimicking mRNA), thereby preventing nuclear Rrp6-dependent trimming. To determine the cellular localization of mature and 5′extended box C/D and H/ACA snoRNA in *rnt1Δ*, we employed fluorescence in situ hybridization (FISH) analysis using fluorescent oligonucleotide probes complementary to mature or 5′extended snoRNA sequence. A probe targeting box C/D snR68 5′extension showed that pre-snR68 was distributed across the whole *rnt1Δ* cell (Fig. 5a). Hybridization with a probe against snR68 mature sequence also showed some cytoplasmic mislocalization of snR68. This indicates that 5′extended box C/D pre-snoRNA are exported from the nucleus in the *rnt1Δ* strain. The control FISH analysis in WT cells revealed that pre- and mature snR68 are localised in the nucleus (Fig. 5a). Note that the probe targeting mature snR68 showed no off-target specificity (Supplementary Fig. 5a). In contrast, precursor and mature RD box H/ACA snR43 and Rnt1-independent box C/D snR13 were exclusively nuclear in *rnt1Δ* cells (Fig. 5b,c). Moreover, box C/D-associated protein Nop1 was normally localized in *rnt1Δ* (Fig. 5d). Note that Rnt1-dependent snoRNA did not display strict nucleolar localization, reflecting the complex maturation of these snoRNA.

We also performed FISH analysis for snR13 in strains where *SNR13* was expressed from the different *GAL1* promoter variants (Fig. 5e and Fig. 4b). In the WT snR13 was localized as nuclear foci. Although snR13 transcribed from the *GAL1* promoter was not restricted to the nucleolus, it was present exclusively in the nucleus. However, the 5′extended snR13e generated in the *GAL1U::SNR13* strain was localized to both nucleus and cytoplasm. Consistent with previous results, the presence of box H/ACA snR3 upstream of snR13 supressed this effect, as snR3-snR13e hybrid was detected only in the nucleus. These reporter construct data reinforce and confirm FISH results for *rnt1Δ* showing that 5′end processing regulates nuclear localization of box C/D snoRNA.

Our cell imaging results demonstrate that 5′end processing of RD box C/D snoRNA precursors is critical for nuclear localization. In effect, Rnt1-dependent cleavage signals the cellular fate of snoRNA versus mRNA.

**5′extended box C/D snoRNA precursors are non-functional**. Efficient expression of box C/D snoRNA is required to achieve full methylation of rRNA. Consequently, if 5′extended box C/D snoRNA is mislocalized away from the nucleolus, this should impede rRNA synthesis. We therefore tested if disruption of snoRNA 5′end processing affects rRNA 2'-O-methylation in *rnt1Δ* strain. To determine the methylation levels of three clusters of methylated nucleotides in 25S rRNA (Fig. 6a), we employed a quantitative RT-PCR approach using low nucleotide concentrations for reverse transcription[41] (RTLN-qP) (Fig. 6b). Low nucleotide concentration causes RT blockage at the sites of RNA methylation[41]. Thus, levels of cDNA synthesised at low dNTP concentration over a particular RNA position, compared to

normal concentration, indirectly measure RNA methylation. In qPCR analyses, the number of PCR cycles (threshold cycle; Ct) required to detect DNA effectively quantitates cDNA levels. In WT, methylation of 25S rRNA delayed Ct from region 2 and 3 by ~2 cycles and for region 4 by almost 7 cycles. In contrast, in *rnt1Δ* Ct for RT-LN products were delayed by only 0.36 cycle for region 2 and 3 and by 1.7 cycle for region 4 (Fig. 6c). Overall, the amounts of cDNA synthesized over methylated nucleotide clusters in low dNTP conditions in *rnt1Δ* were approximately 4–16 times higher than in the WT, indicative of inefficient 25S rRNA methylation in the *rnt1Δ* strain. RTLN-qP analysis of 25S rRNA in *rrp6Δ* cells indicate that the lack of 3′end trimming itself has no effect on box C/D snoRNP enzymatic activity (Supplementary Fig. 6a).

We also tested site-specific 2'-O-methylation levels of 25S and 18S rRNA in the *rnt1Δ* strain using the 8–17 and 10–23 DNAzyme-dependent approaches[42]. DNAzymes are single stranded DNA which form a stem-loop structure around target RNA, cleaving RNA complementary to the junction of the loop arms, either downstream of guanine (8–17 DNAzyme) or between the purine and pyrimidine nucleotides (10–23 DNAzyme). Methylation of the nucleotide located downstream of the cleavage site strongly inhibits DNAzyme activity (Fig. 6d)[42,43]. RNA isolated from *rnt1Δ* strain revealed that DNAzymes targeting snR56- and snR79-site specific methylation in 18S rRNA as well as snR68-site specific methylation in 25S rRNA (Supplementary Fig. 6b) partially cleaved rRNA, generating cleavage products cA and cB (Fig. 6e). RNA from the WT strain was resistant to DNAzyme-dependent cleavage, implying complete methylation of rRNA. To exclude a general defect in rRNA processing we analysed the snR13-specific methylation site of 25S rRNA in the *rnt1Δ* strain. No cleavage products were detected for RNA either in WT or *rnt1Δ* strains, indicating that Rnt1-independent snR13 is unaffected in the *rnt1Δ* strain (Fig. 6e). Overall, these data show that lack of 5′end processing affects RD box C/D snoRNA function. Note that rRNA in the *rnt1Δ* strain is still methylated to some extent as pre-snoRNA can be inefficiently processed at its 5′end by the alternative pathway. Consistently, endonucleolytic maturation of pre-rRNA mediated by U14 snoRNA (snR128)[44], which is processed from pre-snR190-snR128 dicistronic transcript[17], is kinetically delayed in *rnt1Δ*[45]. Moreover, our RNA-seq analysis shows that U14 levels are much higher when compared to other snoRNA in both WT and *rnt1Δ* strains (Supplementary Fig. 6c). We predict that U14 is processed by an alternative pathway in *rnt1Δ* cells, so it can separately, or as a 5′truncated snR190-snR128 dicistron, process pre-rRNA.

We also tested if site-specific snR13-dependent methylation of 25S rRNA was affected in the strain expressing 5′extended snR13e (Fig. 6f). SnR13 guides methylation of two adjacent adenines at positions 2280 and 2281 in 25S rRNA[8]. Since the nucleotide following the second adenine is a pyrimidine (U), we analysed snR13-mediated methylation at this position using the 10–23 DNAzyme (Supplementary Fig. 6d)[42]. SnR13-dependent rRNA methylation in *GAL1::SNR13* strains occurs at the same level as in WT when grown in galactose-containing medium (Supplementary Fig. 6e), so reflecting physiological conditions. The dynamics of snR13-dependent methylation of 25S rRNA during transcriptional induction of normal snR13 and 5′extended snR13e was analysed by Northern blot of RNA samples collected at different time points, following incubation with the DNAzyme targeting snR13-dependent methylation site (Fig. 6f). All 25S rRNA species were detectable by RNA staining on the membrane, while hybridization with a radioactive probe complementary to the region downstream of the cleavage site, detected only full-length 25S and cB product. Prior to *SNR13* induction, in both strains 25S rRNA was almost completely digested by the

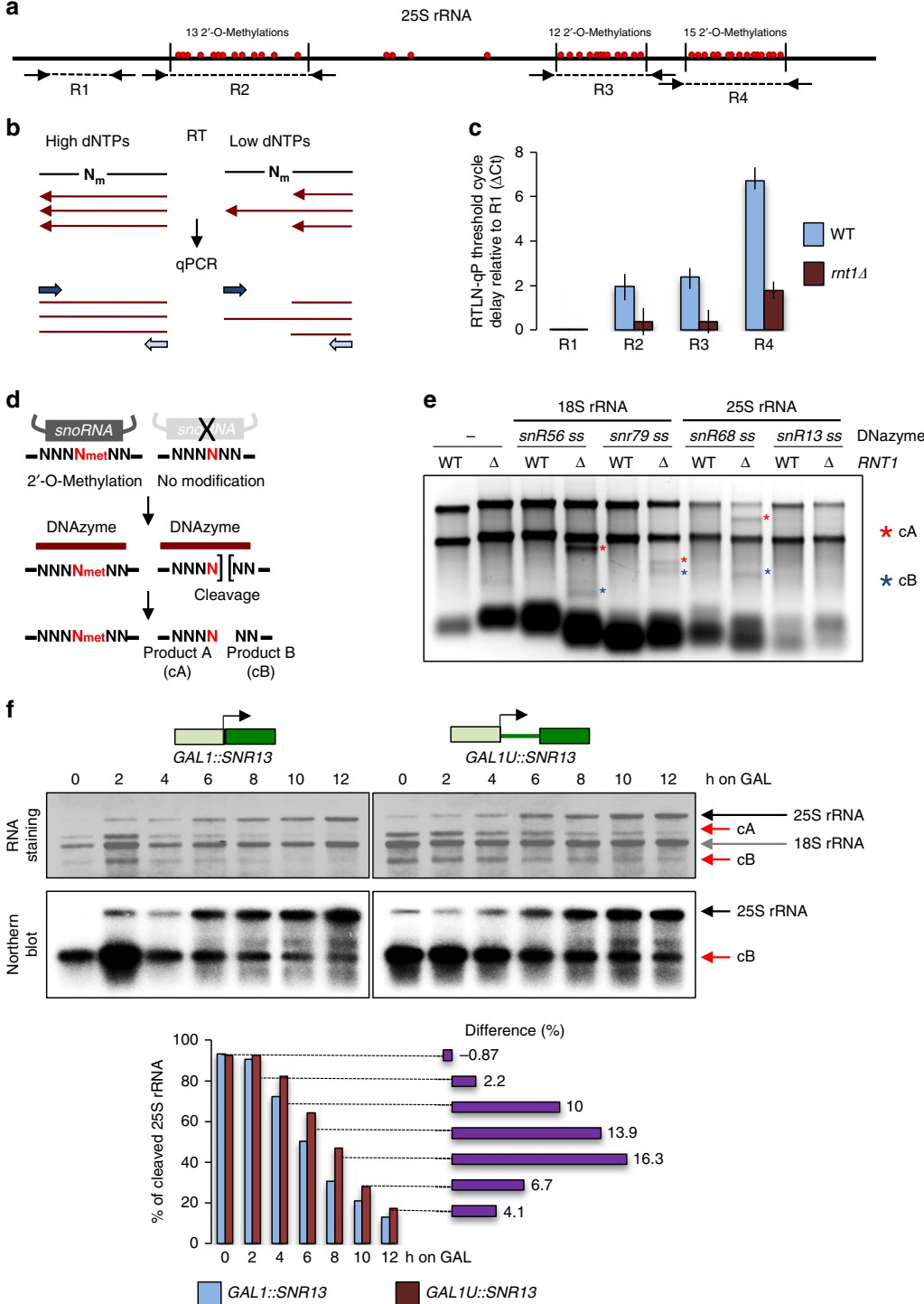

**Fig. 6** Lack of snoRNA 5′processing affects 2'-O-methylation of rRNA. **a** Diagram showing 25S rRNA methylation sites and amplicon locations for RTLN-qP analysis (R1–4). **b** Principles of RTLN-qP technique showing that reverse transcriptase pauses and terminates upstream of modified nucleotide at low dNTP concentration. **c** Delay in RTLN-qP threshold cycle (Ct) of amplicons located over the 25S rRNA methylation clusters in WT versus *rnt1Δ* strains reflecting cDNA levels. The R2–R3 Ct were normalized to the delay of RTLN-qP reaction over non-methylated R1 giving a ΔCt value as graphically presented. Average from three independent biological replicates is shown. Error bars represent standard deviation. **d** Diagram illustrating DNAzyme-dependent cleavage assay. **e** Analysis of site-specific 18S and 25S rRNA methylation using DNAzyme-dependent assay. RNA was visualised by EtBr staining in a denaturing agarose gel. Cleavage products cA and cB are marked by asterisks. **f** Kinetics of snR13-dependent 25S rRNA methylation mediated by WT snR13 and 5′extended snR13e revealed by Northern blot of DNAzyme-dependent cleavage. Upper panels show methylene blue staining of RNA transferred on the membrane while lower panels are Northern blot using oligo probe located downstream of the snR13-dependent methylation sites. Arrows indicate 25S and 18S rRNA positions as well as products of the DNAzyme-dependent cleavage of 25S rRNA (fragments cA and cB). Quantification of the Northern blot indicating percent of cleaved mature 25S rRNA is shown below

DNAzyme targeting the snR13-dependent site, indicating low levels of snR13-dependent rRNA methylation. During transcriptional induction of *SNR13*, 25S rRNA became gradually resistant to the DNAzyme activity, reflecting methylation of newly synthesized 25S rRNA. This process was clearly quantitatively slower in the strain expressing 5'extended snR13e (*GAL1U:: SNR13*) (Fig. 6f). Analysis of snR13 and snR13e levels shows that snR13e did not accumulate as much as WT snR13 in the later time points (0.8× fold lower after 10 and 12 h of induction) (Supplementary Fig. 6f). However, 25S rRNA became equally resistant to DNAzyme cleavage after about 12 h, which suggests sufficient accumulation of functional snR13e. These data demonstrate that snR13e expression induces a specific rRNA methylation defect. We can infer that RD box C/D snoRNA requires 5'end processing to be fully functional.

## Discussion

Pol II generates a wide range of ncRNA which display different cellular fates to mRNA. Mechanisms, which differentiate their RNA metabolism, are crucial to direct their correct maturation pathways. Distinguishing mRNA and ncRNA is often achieved by distinct transcription termination pathway leading either to RNA stabilization or degradation. We describe an additional mechanism for the snoRNA of *S. cerevisiae* which involves their co-transcriptional cleavage by Rnt1. While m[7]G cap is an essential element for mRNA, it is removed from Rnt1 dependent (RD) box C/D pre-snoRNA to facilitate their independent maturation and function.

NNS-dependent transcription termination releases Pol II from non-coding TUs and also recruits enzymatic activities required for their 3'end processing such as the TRAMP complex, which oligoadenylates RNA to stimulate degradation by the nuclear exosome[21,38]. NNS also interacts with the Rnt1 endonuclease[24], implying a further role in Rnt1 recruitment. We show that Rnt1 is co-transcriptionally recruited over snoRNA 3'ends together with NNS, even though it acts at the 5'ends of these transcripts

(Fig. 1c–e). The detection of RNA 3'ends generated by Rnt1 over RCS while associated with transcribing Pol II (Fig. 1g) indicates that Rnt1 cleaves pre-snoRNA co-transcriptionally. Possibly, delayed recruitment of Rnt1 to snoRNA 3'ends ensures that they remain unprocessed at their 5'ends until snoRNP assembly occurs. NNS also co-purifies with the CBC[21]. However, CBC is recruited to snoRNA genes at an early stage of transcription (Fig. 1c–e) as occurs for protein-coding genes[26]. CBC–NNS interactions may play post-transcriptional roles by recruiting the exosome to trigger nuclear RNA degradation[37]. The pausing of Pol II over NNS-dependent terminators[12] may create a time window prior to RNA release from the DNA template. This allows snoRNP assembly and recruitment of factors required for subsequent maturation. Therefore, snoRNA terminators emerge as processing hubs that not only mediate transcription termination but also define this class of Pol II-dependent transcript as snoRNA.

Our data indicate that the removal of m[7]G cap is pivotal to inform the transcription machinery that the nascent RNA is not mRNA, so committing it to the RD box C/D snoRNA maturation pathway. Consequently, Rnt1-dependent cleavage at the 5'ends of box C/D pre-snoRNA regulates 3'end processing and nuclear retention (Figs 3 and 5). In effect, the presence of cap (or associated proteins) emerges as a checkpoint in RD snoRNA synthesis. We hypothesise that m[7]G may compete with box C/D snoRNP proteins to signal the transcription machinery to direct the transcript to either mRNA or snoRNA maturation pathways (Fig. 7).

In the case of box C/D snoRNA where the TSS is adjacent to the mature snoRNA sequence (snR13, 4, 45 and 17), immediate association of snoRNP proteins may prevent conformational changes in the Pol II complex mediated by cap, which would otherwise classify the nascent transcript as mRNA. Therefore, these snoRNA remain capped at their 5'ends[15]. In all cases the presence of m[7]G had no effect on 3'end processing. Consistently, snR13 artificially extended at the 5'end with the *GAL1* UTR (snR13e) was unprocessed at the 3'end while snR13 transcribed from the *GAL1* promoter lacking the UTR was normally

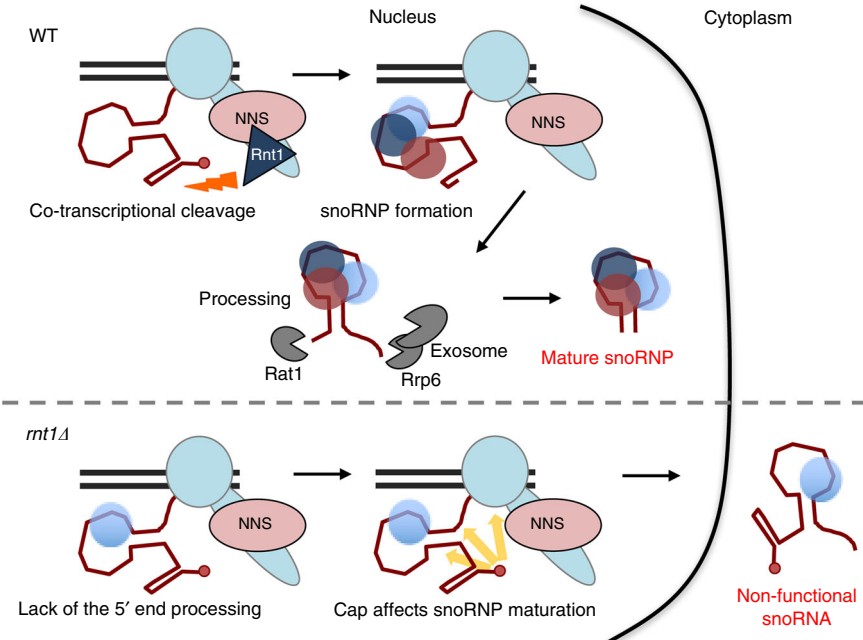

**Fig. 7** Model for box C/D snoRNA maturation in yeast. Co-transcriptional cleavage mediated by Rnt1 removes the cap structure from box C/D pre-snoRNA and so directs the precursor to the snoRNP synthesis pathway in WT cells. In the *rnt1Δ* strain, retained m[7]G cap marks the RD box C/D pre-snoRNA as mRNA resulting in cytoplasmic localization of 5' and 3'unprocessed snoRNA

processed (Fig. 4b). In contrast to most C/D snoRNA, 5′caps are essential for box H/ACA snoRNA synthesis (Fig. 2b,e) and furthermore their 3′end processing is unaffected by cap retention (Fig. 3b). This reflects the genomic organization of box H/ACA snoRNA where Rnt1-dependent 5′end processing is rare and their 5′ends are defined by transcription initiation with the cap structure remaining during subsequent processing. However, mature capped box H/ACA and Rnt1-indpendent box C/D snoRNA have an altered cap structure[15] modified by Tgs1, which converts $m^7G$ cap into trimethylated cap (TMG)[20]. It is plausible that Tgs1 modifies cap in order to disturb $m^7G$-CBC interactions[46]. Finally, our studies on yeast snoRNA synthesis indicate that CBC is not required for this process even though CBC was previously shown to enhance transcription termination and processing of human snRNA[22,23].

It is informative to consider the evolution of snoRNA gene arrangement and prevalence in eukaryotes. Notably, snoRNA numbers increase with organism complexity. *S. cerevisiae* has 77 while *Drosophila melanogaster* has 227 snoRNAs. Remarkably humans are estimated to possess 450–700 snoRNA genes [47,48]. Conversely, the fraction of independently transcribed snoRNA decreases with organism complexity, being replaced by either polycistronic TUs or intronic localization. In *S. cerevisiae*, only 10% of 77 snoRNA are located within the introns of protein coding genes[7,8], while in the human genome most snoRNA are intronic[7,47,48]. Such an intronic localization of snoRNA may have evolved to facilitate co-expression with host ribosomal protein coding genes[47,49]. However, host genes for snoRNA appear to correlate with expression levels rather than gene ontology. In general, snoRNA that modify abundant rRNA are located in the introns of highly transcribed protein-coding genes, so providing high expression levels[50]. Why evolution has selected against independently transcribed snoRNA genes in higher eukaryotes and led to a loss of dedicated snoRNA promoters remains unknown. We show that the cap structure may be used to distinguish box C/D snoRNA from mRNA and so specify correct maturation pathways. Possibly, this may have contributed to the evolutionary pressure to remove independent transcription initiation sites for many snoRNA. Since $m^7G$ cap does not affect box H/ACA snoRNA maturation, apparently cap removal may be only one of several ways to shape genomic organization and processing pathways of snoRNA genes in higher organisms.

## Methods

**Yeast strains construction**. Yeast strains used in this work are listed in Supplementary Table 4. The transformation procedure was as described[51] using standard lithium acetate method. Strains were generated by a one-step PCR procedure[52]. To construct strains expressing snoRNA from an inducible *GAL1* promoter, the region of the *GAL1* promoter was amplified by PCR using the *pFA6a-KanMx6-pGAL1* plasmid. The *GAL1*-snoRNA modules were then further transferred between strains by amplification of the *GAL1*-snoRNA cassette on the genomic DNA template followed by transformation into yeast.

**Yeast growth conditions**. Strains were grown at 23 °C or 30 °C in YPD medium (1% yeast extract, 2% Bacto-peptone, 2% glucose) to mid-exponential phase. Strains containing conditional temperature-sensitive alleles were pre-grown at 23 ° C, up to mid-exponential phase and transferred to 37 °C. Transcription from *GAL1* promoters was induced by addition of 2% galactose to yeast cultures pre-grown in minimal SC (0.67% yeast nitrogen base, supplemented with required amount of amino acids and nucleotide bases) containing 2% raffinose and 0.08% glucose. Each experiment was biologically replicated unless otherwise stated.

**RNA methods**. Total RNA from yeast cells was isolated using a hot phenol procedure[53]. Northern hybridization was essentially as described[54]. 8 µg of total RNA was separated on 6% denaturing polyacrylamide-urea gels, electro-transferred (TransBlot Biorad, 100 mA for 45 min) onto nylon membranes (GE Healthcare) and hybridized using PerfectHyb buffer (Sigma) with oligonucleotides labelled with $^{32}P$ at their 5′ends. Overnight hybridization at 42 °C was followed by three washes with 6xSSPE. Hybridization signals were visualized and quantified using FLA5000 imaging system (Fuji). In most cases images were cropped on their edges in

Photoshop. Raw, uncropped images from main figures are shown in Supplementary Fig. 7. Oligonucleotide probes used for hybridizations are listed in Supplementary Table 5. RNase H treatment was as described previously[14], briefly 10 mg of RNA in 1 × RNase H buffer was incubated with 1 pmol of oligo for 10 min at 65 ° C and cooled down to 30 °C. Next 10 U of RNaseH (NEB) was added and RNA was digested for 1 h. For RNA-seq, 5 µg of total RNA was rRNA-depleted using Ribo-Zero kit from Illumina. The so obtained rRNA-depleted fraction (100–500 ng) was used to prepare libraries employing Ion Total RNA-seq Kit v2 (Thermo Fisher) and subsequently sequenced using the Ion Proton system.

**ChIP and ChIP-seq**. Chromatin was precipitated as previously described[14]. 100 ml of culture ($OD_{600} = 0.4$–0.8) was crosslinked with 1% (v/v) formaldehyde at room temperature for 20 min and quenched with 375 mM glycine for 5 min. Cells were resuspended in 1 ml of cold FA1-lysis buffer (50 mM HEPES-KOH pH 7.5, 150 mM NaCl, 1 mM EDTA, 1% Triton X-100, 0.1% sodium deoxycholate, protease inhibitors (Complete, Roche)) and disrupted with 300 µl of zirconia beads using a MagnaLyser (three times for 30 s at maximum speed with 5 min rest period on ice between runs). The lysate was diluted with 1 ml of FA1-lysis buffer and sonicated in a Bioruptor sonicator (Diagenode) for 15 min (15 s on, 15 s off) set at medium level. The lysate was clarified by 40 min spin at $16000 \times g$ at 4 °C. 500 µl of the extract was diluted 5 times with FA-1 buffer and 1 ml was incubated overnight at 4 °C with 10 µl anti-Myc (ab9132, Abcam) or 1 µl anti-Rbp3 (1Y26, Neoclone) antibody. Next extract was incubated with a 1:1 mix of 60 µl of Dynabeads Protein G and A (Invitrogen) for 2 h at 4 °C. Beads were washed four times at room temperature with 1 ml of FA1-lysis buffer and once with 1 ml FA2-lysis buffer (50 mM HEPES-KOH pH 7.5, 500 mM NaCl, 1 mM EDTA, 1% Triton X-100, and 0.1% sodium deoxycholate), ChIP wash buffer (10 mM Tris-HCl pH 8.0, 250 mM LiCl, 1 mM EDTA, 0.5% Nonidet P-40, 0.5% sodium deoxycholate) and TE (10 mM Tris-HCl pH 8.0, 1 mM EDTA). Beads were resuspended in 100 µl of ChIP elution buffer (50 mM Tris-HCl pH 7.5, 10 mM EDTA, 1% SDS) and samples, including 20 µl of the input sample, were incubated with 40 µg of Proteinase K (Bioline) for 2 h at 50 °C and 6 h at 65 °C. For RNase treatment the diluted extract was incubated with or without RNase A (10 U, Qiagen) and T1 (500 U, Roche) at 37 °C for 1 h. Extracts were then incubated overnight at 4 °C, then for 2 h with 4 µl anti-Myc (ab9132, Abcam) or 1 µl anti-Rbp3 (1Y26, Neoclone) antibody. Next the extract was incubated with a 1:1 mix of 25 µl of Dynabeads Protein G and A (Invitrogen) for 2 h at 4 °C. Washes were performed as above. Beads resuspended in elution buffer with Proteinase K were incubated 2 h at 56 °C and 12 h at 65 °C. DNA was purified using the commercial clean-up kit (Qiagen). For ChIP-seq analysis at least four IPs were pooled and used for subsequent treatment. The libraries for ChIP-seq were prepared using NEBNext ChIP-seq Library Prep (NEB) and sequenced on Illumina HiSeq400 by the High Throughput Genomics Group, University of Oxford. ChIP followed by qPCR analyses are shown as an average of three independent biological replicates.

**Bioinformatics methods**. *S. cerevisiae* genome, scaCer3 (April 2011) was downloaded from UCSC (http://hgdownload-test.cse.ucsc.edu/goldenPath/sacCer3). Gene boundaries were obtained from the Saccharomyces Genome Database (SGD, http://www.yeastgenome.org/) for the same version.

Mapping RNA-seq sequencing reads: For Ion Torrent RNA-seq sequencing, single-end reads were mapped to sacCer3 genome using two-step alignments. First, the reads were aligned with TopHat[55]. Second, the resulting unmapped reads from the first step were extracted and aligned with Bowtie2[56] with --very-sensitive-local and --local options. Uniquely mapped reads with no mismatches and mapping quality ≥30 from both steps were then combined using SAMtools merge[57]. Number of reads mapped to each gene was normalized to its length and total number of genome-aligned reads (RPKM value, Reads Per Kilobase of exon model per Million mapped reads). The bigWig tracks from the resulting normalized samples were visualized in a custom UCSC Genome Browser track data hub, hosting the sacCer3 reference genome.

ChIP-seq data analysis: For Chip-seq, paired-end reads for each sample were mapped to the sacCer3 genome (UCSC, downloaded from http://hgdownload-test.cse.ucsc.edu/goldenPath/sacCer3) using the Bowtie2 alignment software. Uniquely mapped reads with a mapping quality ≥30 were retained for further analysis. Peaks were called using Model-based Analysis, MACS2[58] for ChIP-Seq with default options. Only those peaks with q-value below 0.05 were retained for further analysis. This resulted in 1238 peaks for Rnt1 and 3801 peaks for Cbp20.

Data visualization: Metagenes showing 3′end of Rnt1-dependent box C/D snoRNA and Rnt1-independent box C/D and H/ACA snoRNA was generated by plotting normalized read counts around annotated 3′end for sense strand relative to the direction of gene transcription. For box H/ACA snoRNA metagene snR30 and snR35 were discarded due to their very high signals relative to the other box H/ACA snoRNA. Next average reads from *rnt1Δ* were normalised to the average reads of the last 6 nucleotides from the coding sequence in WT by a factor 1.73. NET-seq data for wildtype yeast BY4741 was downloaded from GSM617027. Normalized read counts were calculated for sense and antisense strands, relative to the direction of gene transcription for a region of 25 bp upstream and downstream of annotated AGNN positions and plotted. SnoRNA used for metagene analysis are listed in Supplementary Table 3.

**Circular RNA RT-PCR analysis**. If required 5′cap was removed by RNAse H treatment in the presence of specific oligonucleotide. Next 10 µg of total RNA was circularized using 30 U of T4 RNA ligase (NEB) in 37 °C for 1 h in total volume of 30 µl. CR-RNA was purified by phenol followed chloroform extraction, precipitated and suspended in 10 µl H$_2$O. 1 µl of RNA was used for cDNA synthesis (AMV, Promega). Primers used for reverse transcription and PCR are listed in Supplementary Table 5 (primers for RT are marked as "RT"). PCR products were purified and then cloned into pGEM easy vector (Promega). Isolated clones were sequenced using T7 primers.

**Fluorescent in situ hybridization**. Strains were grown in YPD medium at 25 °C to log phase. Cells in 10 ml of medium were prefixed with 37% formaldehyde (final concentration 4%) for 15 min and harvested by spinning down. Cells were fixed in 5 ml of solution A (4% paraformaldehyde, 0,1 M KPO$_4$ (pH 6,5), and 5 mM MgCl$_2$) for 3 h. Next cells were washed twice with solution B (1,2 M sorbitol and 0,1 M KPO$_4$, pH 6,5), resuspended in 0,5 ml of solution B with 0,05% β-mercaptoethanol and freshly prepared Lyticase (Sigma Aldrich), digestion was performed at 37 °C for 20 min. Cells were washed tree times in ice cold solution B, and then resuspended in 0,3 ml of solution B and stored overnight at 4 °C. Spheroplasts were plated to the wells of white glass slides printed with Epoxy (Thermo Scientific™ Diagnostic Slides) that had been covered with a 0,1% poly-L-lysine-containing solution (Sigma Aldrich). Cells were washed with 70, 90 and 100% ethanol for 5 min respectively. Pre-hybridization was conducted in humid chamber for 2 h at 37 °C in buffer containing 10% dextran sulphate, 0,2% BSA, 2 × SSC), 125 µg Escherichia coli tRNA/ ml, 0.5 mg/ml single—strand DNA denatured (95 °C for 3 min.) and Ribolock 1 U/ul (Thermo Scientific). The same buffer was used for hybridization with probes at final concentration 100 pg/ml. 30 nt long probes (listed in Supplementary Table 5) were labelled at their 5′end with fluorescent dye Alexa fluor 488 and Alexa fluor 647 (Sigma Aldrich). Hybridization was performed in humid chamber overnight at 37 °C. After hybridization cells were washed three times for 10 min with 2 × SSC at 37 °C and three times with 1 × SSC at room temperature. Cells were briefly washed with 4 × SSC containing 1% Triton X-100 followed by two more washes with 4 × SSC, each wash lasting for 10 min. Nuclei were stained with 0.1 µg/ml DAPI in 1 × PBS. Slides were then mounted with VECTRASHIELD Mounting Medium (Vector laboratories) and stored at −20 °C. Images were acquired using Carl Zeiss Axio Imager Z2 confocal microscope with 63 × NA 1.4 oil objective and Zen software. 3D datasets were generated by multiple 200 nm z stacks covering the entire cell volume. 2D datasets were obtained in ImageJ by maximum projection function.

**rRNA methylation analyses**. Reverse Transcription at Low deoxyribonucleoside triphosphate concentrations followed by quantitative polymerase chain reaction (RTLN-qP) was performed as described previously[41] with some modifications. Briefly 1 µg of total DNAse I treated RNA was incubated at 70 °C with 250 ng of random hexamers, chilled on ice and incubated at 25 °C for 5 min. Reverse transcription was performed using 20 U of AMV reverse transcriptase (Promega) in 1 mM (high concentration) or 5 µM (low concentration) of dNTPs at 42 °C for 1 h. This was followed by qPCR analysis using Sensimix master mix (Bioline). Oligonucleotides are listed in Supplementary Table 5.

**Calculation of reverse transcription in low nucleotides concentration followed by qPCR values**. In the RTLN-qP analysis we calculated how many cycles later the Ct is determined for the product of RT reaction performed in low dNTPs concentration (5 µM) as compared to the Ct for the product of RT in normal dNTPs concentration (1 mM). The Ct delay value (ΔCt) was determined for each cluster of methylated nucleotides in 25S rRNA (regions 2–4, R2–4) and for the region located in 25S 5′end, which is unmethylated (R1) (Fig. 6a). As cDNA for all regions was synthesized in the same reaction, we used ΔCt for region 1 to define RT efficiency for a particular reaction. To visualize how RNA methylation suppressed cDNA synthesis in low dNTPs concentration, ΔCt for the non-methylated region 1 was deducted from ΔCt obtained for regions 2–4.

**DNAzyme-dependent assay**. DNAzyme dependent analysis was performed as described[42]. For 10–23 DNAzyme treatment 5 µg of DNase I treated RNA was combined with 200 pmol of 10–23 DNAzyme and 2.5 µl of incubation buffer (4 × concentrated: 24 mM Tris pH 8; 60 mM NaCl) in a final 10 µl volume. After heating at 95 °C for 3 min, the reaction was placed on ice for 5 min. Next 1 µl of Ribolock (Fermentas) was added and reaction was incubation at 25 °C for 10 min. Temperature was then raised to 37 °C and 4 µl of pre-warmed reaction buffer (4 × concentrated: 200 mM Tris pH 8; 600 mM NaCl) and 4 µl of 300 mM MgCl$_2$ was added. Finally, pre-warmed water was added to a final volume 20 µl. The reaction was continued for 1 hr at 37 °C. For reactions with 8–17 DNAzymes, 5 µg of DNAse I treated RNA and 400 pmol of DNAzyme (each in a volume of 4 µl) were heated separately at 95 °C for 2 min following by incubation at 25 °C for 10 min. 8 µl of reaction buffer (2 × concentrated: 200 mM KCl, 800 mM NaCl, 100 mM HEPES pH 7.0, 15 mM MgCl$_2$, and 15 mM MnCl$_2$) was then added to the DNAzyme. RNA and DNAzyme were mixed and incubated at 25 °C for 2 h. h. After incubation with either 10–23 or 8–17 DNAzyme, RNA was extracted with phenol/chloroform, precipitated, separated on 1% agaose-formaldehyde gel and followed by Northern Blot analysis if necessary. Each experiment was replicated two times. Oligonucleotides are listed in Supplementary Table 5. Raw, uncropped images from main figures are shown in Supplementary Fig. 7.

**Data availability**. GEO accession number: GSE93240. All data are provided by the authors upon reasonable request.

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

## Acknowledgements

We thank A. Hinnebusch for *CBP20-myc* and *CBP80-myc*, G. Chanfreau for *rnt1E320K* and T.H. Jensen for *ceg1-63* strains. The JK group was supported by the Polish Ministry of Science and Higher Education (N N301 065740 to J.K.), Polish-Swiss Research Programme (PSPB-183/2010 to JK) National Science Centre (UMO-2011/01/N/NZ1/04344 to SS) and EU Social Fund (UDAPOKL.04.01.01-00-072/09-00). The JK group used CePT infrastructure financed by the EU Regional Development Fund (Innovative economy 2007-13, POIG.02.02.00-14-024/08-00). The PG group is supported by a Sir Henry Dale Fellowship jointly funded by the Wellcome Trust and the Royal Society (200473/Z/16/Z). The NJP laboratory was supported by a Wellcome Trust Programme Grant (091805/Z/10/Z) and European Research Council Advanced Grant (339270). We thank the High-Throughput Genomics Group at the Wellcome Trust Centre for Human Genetics (funded by Wellcome Trust grant reference 203141/Z/16/Z) for the generation of the Sequencing data.

## Author contributions

P.G., S.A.S., J.K. and N.J.P. designed experiments. P.G. and S.A.S. performed the majority of experiments. S.D. carried out the bioinformatics analyses, A.P. performed the FISH analysis, Z.M. assisted with strain construction and Northern Blot analyses while HP assisted with rRNA methylation analyses. H.E.M. advised on data analysis and experimental design. P.G., J.K. and N.J.P. wrote the manuscript.

## Additional information

**Competing interests:** The authors declare no competing interests.

