## [Peer Review File · Nature Communications]

Reviewers' Comments:

Reviewer #1:

Remarks to the Author:

Previous analyses reported that most mature, independently transcribed, C/D snoRNAs tested lack TMG caps and to show 5' maturation that is variably dependent on cotranscriptional cleavage by the Rnt1 endonuclease or 5' digestion by the Xrn1 and Rat1 exonucleases. This was also reported to be the case for a smaller subset of H/ACA snoRNAs. The authors present evidence for 5' cleavage of a small number of additional C/D and H/ACA snoRNAs, based on 5' extensions in *rnt1Δ* strains. Circular PCR indicated that 5' extended pre-snoRNAs are also 3' immature. The authors conclude that this demonstrates that 5' processing precedes, and may be required for, 3' processing of C/D snoRNAs. This is supported by cis-acting mutation in SNR68 and catalytic mutation in Rnt1. The lack of capping activity reduced the impairment of 3' processing in a GAL-SNR13 fusion construct suggesting that presence of the cap might delay 3' processing. Further data indicate that capped pre-snoRNAs can partially leak to the cytoplasm and appear to be defective in Nop1 binding. Consistent with these findings, rRNA methyltransferase activity was also reduced.

Overall, there is a tendency to draw strong general conclusions from the behavior of individual, artificial constructs. However, the work makes a useful contribution to understanding snoRNA maturation and will be of interest and significance to the field. There are some points that should be addressed.

Specific comments

1) The ChIP data in Figure 1 are presented in the context of discussion of the role of Rnt1 in 5' processing, but would appear to be more consistent with its 3' function. There does not seem to be an obvious difference between snoRNAs predicted to show or lack 5' cleavage. Are there any systematic differences between the Rnt1 or CBC distribution on snoRNAs with predicted 5' Rnt1 cleavage sites and snoRNAs or snRNAs that lack 5' cleavage?

2) Pre-snoRNAs stabilized by loss of Rnt1 were previously reported to carry TMG caps (Lee et al 2003). In this work, the pre-snoRNAs are reported to carry m7G caps in the Abstract but this does not appear to have been addressed experimentally. This is important because TMG caps are not expected to be bound by CBC. It would therefore be expected that CBC would be displaced from the pre-snoRNA prior to or at cotranscriptional hypermethylation. This would be very relevant to the ChIP data for CBC binding. Reduced CBC signal is interpreted as reflecting cleavage, but could also reflect hypermethylation. This point should be addressed.

3) Previous analyses indicated that the level of 5' Rnt1 cleavage was highly variable between snoRNA species. This should be discussed with respect to the data presented here.

4) P8: The authors refer to "3'extended, Rrp6-dependent pre-snR13 precursors".

By this do the authors intend:

"3'extended, pre-snR13 that accumulates in the absence of Rrp6"?

The, presumably corresponding, snR13 Me band in the northern is described in the figure legend as "precursors as with short oligoadenylated 3'extensions". How was this shown?

5) P9: "Overall, we conclude that the synthesis of box H/ACA snoRNA requires m7G cap. In contrast this structure is dispensable for box C/D snoRNA expression...."

This seems to be a remarkably sweeping statement based on two artificial constructs.

6) Figure 6e: It would be important to show the relative expression of snR13 from the GAL1 and GAL1U constructs.

7) Since the 5' extended pre-snoRNAs are also 3' extended, is it clear that the 5' extensions rather than 3' extensions are responsible for the observed defects?

Reviewer #2:

Remarks to the Author:

This is a nice study. The authors utilize a broad set of complementary tools to interrogate this aspect of snoRNA function, and the largely systematic and well-conceived experiments support the model presented in Figure 7. While I find the work significantly impressive, this is not my field and I am thus not qualified to comment on the potential impact of this work. Potential issues that might be addressed to improve the quality of the manuscript are listed below:

(1) While some of it had to do with my lack of expertise with this field, the difficulty that I had in reading this manuscript was also due to less-than-ideal editing and organization. It seems that approaches could be rationalized a bit more cleanly and some degree of discussion surrounding each of the drawn conclusions in the results section would be better presented in the discussion proper. While this latter point may only be my personal preference, I found the speculation around the conclusions disruptive to the otherwise strong logic flow in the work. Examples include but are not limited to lines 203-207 and 238-241 in the manuscript.

(2) For figure 1, the logical argument presented in support of the RNA-independence of Rnt1 recruitment (lines 134-139) is reasonable, but direct demonstration of this would certainly strengthen the position. Perhaps this has already been directly demonstrated elsewhere?

(3) Also regarding the data within Figure 1 - While the conclusions drawn here (lines 162-165) are generally supported by the data, I didn't quite follow the RNA stem-loop independence assertion. Perhaps this could be better described or phrased for the non-expert?

(4) Figure 2 - The methodological description of the *ceg1-63* induction experiments here is a bit unclear. Were the WT controls also assayed at the restrictive temperature? Is processing any different in the *ceg1-63* line at the permissive temperature? I am sure that the possibility that these observations are an artefact of temperature or strain were ruled out, but it would be an improvement if this were explicitly explained for the audience.

(5) Also regarding the data within Figure 2: The conclusion drawn from the data within this Figure (lines 203-207) is that synthesis of box H/ACA snoRNAs requires a cap, and while structure is dispensable for box C/D snoRNA expression, where absence of the cap accelerates 3' end processing. Again, perhaps demonstrating my ignorance of this field, I am not sure that this conclusion can be broadly generalized as it appears to be here, particularly given that mutants not dependent on Rnt1 were specifically chosen to be employed. This concern can actually be extended through much of the manuscript – several different box C/D and box H/ACA snRNAs to utilized to make specific points. Perhaps like here (where I actually understood the rationale) this is very careful selection to best illustrate selected points due to specific snRNA sequence or structure (I didn't find rationalizations elsewhere). Perhaps it was a function of what on hand. This strategy supports the assertion that several of the snoRNAs conform to the ultimate model, but that distinct snoRNAs are utilized to test the various aspects of the mechanism challenges the generalizability of the model at the same time. This is a potential weakness of the work – a strategy of consistently sticking with two, say snR68 and snR65, unless absolutely necessary throughout the manuscript would have been preferable.

(6) Conclusion for data described in Figure 3: "Overall these data show a clear interplay. . .and exosome-dependent processing for box C/D snoRNA (lines 255-257)." Any role for the exosome may be reasonably assumed, but was not explicitly demonstrated here. These set of data and the

overall remainder of the conclusion are OK.

(7) Figure 4: This was really systematic, good job. What the GAL1 5' UTR experiments were explicitly testing and what the conclusion drawn from these data could be clarified and better integrated to the overall narrative. I'm not sure that I entirely followed that aspect of the logic here.

(8) Figure 5: The authors conclude that cap removal is critical for nuclear localization and snRNP assembly. This summary can be deduced from the previous data, but that's not what was directly tested here. If this specific conclusion is to be drawn, the finding would be strengthened by repetition of key experiments in *ceg1-63* and *cbp80Δ* mutants.

(9) It seems that the last section of the discussion, which focuses on snoRNA genomic organization as a function of maturation mechanism during evolution, would I be strengthened by an explicit comparison of box C/D and box H/ACA organization from cerevisiae to sapiens. This seems to be the test of the idea put forth – wouldn't one expect that more box C/D than box H/ACA have wound up in introns due to the selective pressure exerted by the mechanisms described here on the former? If this isn't true, I'm not sure that I would make the argument at all - the H/ACA caveat at the end doesn't seem to salvage it. Beyond this, the model here was cerevisiae, and so perhaps the mechanism within human cells should be evaluated prior to drawing these parallels.

Reviewer #3:

Remarks to the Author:

In this manuscript, Grzechnik and co-authors, have taken a global approach to study the 5' processing of snoRNAs and its importance to snoRNA accumulation and function. The authors conclude that 5' processing is important for snoRNA processing and function and propose a mechanism by which Rnt1 cleavage 5' of the snoRNA, which removes the m7G cap structure which is added co-transcriptionally, is needed for efficient snoRNA processing, association with snoRNP proteins, correct localisation and snoRNP function. This is, in principle, an interesting story. However, this manuscript comes across as a rushed first draft and I feel that the way this manuscript has been prepared has made it difficult to judge the data. A significant part of the manuscript focuses on Rnt1 cleavage of pre-snoRNAs yet little is done to explain which elements of this paper are new and which confirm earlier work on Rnt1 and snoRNA processing. Furthermore, the authors make sweeping statements about box C/D snoRNA processing and function based on their data when, in fact, the phenomenon they describe are only seen with a subset of snoRNAs, even if they are the majority.

Major Points:

1) The authors do not clearly explain in the introduction the different snoRNA gene types and which are found in humans (primarily intronic), plants (intronic and polycistronic), and yeast (mainly mono-cistronic with some intronic and some polycistronic). Furthermore, authors make global sweeping statements about yeast snoRNAs, such as the claim that mature yeast box C/D snoRNAs lack a cap (page 3, line 69). This is clearly not the case as 4 mature snoRNAs, U3, snR13, snR4 and snR45 (Balakin et al., Cell 1996) have 5' cap structures. Indeed, the authors manage to contradict themselves by stating this themselves in the results section. Furthermore, one yeast snoRNA, snR52 (which the authors analyse in the manuscript), is transcribed by RNA pol III, a point missed by the authors. The introduction therefore needs to be re-written so that these points are clear to the readers.

2) Throughout the manuscript, the authors make sweeping statements about box C/D snoRNPs that are not true and need explaining properly. The observation that correct 5' processing by Rnt1 is required for snoRNP formation and function only applies to a subset of the box C/D snoRNAs, even if it is the major group, and the text needs to be adjusted to reflect this.

3) Page 5, line 112. The authors claim that 3 snoRNAs retain their cap structures. However, 4 snoRNAs, U3, snR13, snR4 and snR45, have been shown to be capped with an m³G cap (Balakin et al., Cell 1996). Why is U3 omitted from table 1 and not discussed at all in the manuscript? U3 is assembled and processed by the same machinery, as far as we are aware, as the other snoRNAs so this omission makes no sense at all.

4) Figure 2A. The authors claim that Mt represents decapped, 5' truncated transcripts. However, these transcripts are present even before induction of gene expression (time-point 0) and, for the wild-type, do not vary upon induction of gene expression. The authors claim that this is due to basal activity of the gal promoter. If this is the case, why is only the mt form produced, this makes no sense. This observation must be explained by the authors and for me to believe that these are really snR13 transcripts, and not just some cross-reaction of the probe used. Furthermore, if these are snR13 transcripts, I would also need to see whether these transcripts occur at significant levels when snR13 is transcribed from its normal promoter and that this is not some artefact of the Gal1 system used in this experiment. The authors claim that there is only a slight decrease in mature snR13 levels in the ceg1-63 strain. I would need to see quantitation of this experiment before I would believe this. Furthermore, since the authors claim that alternative 3' processing phenotype was seen in the ceg1-63 strain, it would be good to know whether this process is dependent on Rrp6. In Figure S2A the authors used RNase H treatment to characterise the transcripts. Why in this experiment is the Mt form missing? Finally, the authors conclude (page 9, line 203) that the cap structure is dispensable for box C/D snoRNA expression. However, all the data presented for the observation in the paper and the conclusions drawn are based on one snoRNA from each class of snoRNA. It is also possible that the snoRNAs that undergo 5' Rnt1 processing are dependent on the m⁷G cap – a point that should be tested or, if the authors have data on this, discussed in the manuscript.

5) Figure 3. The authors explain that “box C/D snoRNA transcribed with 5' extensions as well as the last snoRNA from polycistronic TUs have short unprocessed extensions at their 3' ends”. The data needed to back up this statement is not presented. Indeed, data is presented for snR51 (I assume, the labels are insufficient to determine whether this is snR51 or the other members of the cluster) but not the rest of the cluster. The authors need to present more data to validate this statement.

6) Page 13, line 320. “Overall, our analysis indicate that the presence of the m⁷G cap but not the associated CBC on the box C/D snoRNA precursor interferes with the final step of 3' processing.” While this is fine the authors do not really propose why this is not a problem with the 4 box C/D snoRNAs that retain the 5' cap structure. The same statement is also made on page 15, line 361. The authors must clarify that this only applies to a subset of the snoRNAs, even if it is the majority, and not all box C/D snoRNAs.

7) Figure 5. Mature snR68 should be in the nucleolus, a cap-like structure within the nucleus, that does not stain well with DAPI. However, the mature snR68 is found throughout the nucleus in Figure 5A. Indeed, the same is seen for snR43 and snR13 (panel c). In contrast, snR13 in panel f, and Nop1 (panel d), a protein associated with both snR13 and snR68, are found in nucleoli and are present a cap at the edge of the DAPI stained region. Unfortunately, in addition I cannot see the differences the authors describe in the text in panel f. The GAL1::SNR13 and the GAL1U::SNR3::SNR13 snoRNAs do not localise the same as the WT snR13, even though they should for the experiment to be valid. I am therefore not satisfied with the quality of the FISH data and find it difficult to draw any conclusions from the experiments. In the IP experiment (panel e), the authors claim that Nop1 was only associated with snoRNAs that were shortened at their 5' ends. However, while the pre-snoRNAs that were shortened at their 5' ends were enriched in the Nop1 IP material, the longer forms were also present and the data is not as “black and white” as the authors describe in the text. Furthermore, I think that the binding of Nop58 to the snoRNAs/pre-snoRNAs needs to be tested as this protein is essential for snoRNA formation and

binds directly to the snoRNA. In contrast, structural data suggest that Nop1 primarily associates with the snoRNAs through its interaction with Nop56 and Nop58.

8) Figure 6a, b and c. The major issue I have with the work presented in this figure is that Rnt1 is important for rRNA processing. This point is not mentioned in the manuscript, which is surprising given the work performed earlier by Kufel on the subject. Lack of Rnt1 has an impact on ribosome biogenesis that could therefore non-specifically affect the results presented in this figure irrespective of Rnt1's role in snoRNA maturation. These points must be acknowledged in the text and controls are needed to verify the specificity of the observations presented. Indeed, in Kufel et al., (RNA, 1999) it was shown that 25S production was reduced in the absence of Rnt1. It might therefore make more sense to look at methylation in the 18S rRNA. The authors have indicated that there are three types of box C/D snoRNA; capped (e.g. snR13), those that are Rnt1 – dependent in 5' processing and those that are Rnt1 - independent (a decent table emphasising this point would be appreciated). Therefore, loss of Rnt1 should only affect those that require Rnt1 for the correct 5' end processing of the snoRNA. In order to demonstrate that only these snoRNAs are affected the authors must show methylation data for individual snoRNAs of each type in control cells and cells lacking Rnt1. This could be achieved through a high-throughput sequencing approach but I would be happy if the authors would use the RNase H approach, established by the Steitz lab, to analyse the activity a couple of examples of Rnt1-dependent and –independent snoRNAs. I would also like to see the impact of Rnt1 on U14 levels and methylation activity. U14 is required for both rRNA methylation and 18S rRNA processing. If U14 is significantly impacted by Rnt1 loss then both rRNA methylation and 18S rRNA processing should be affected. However, as reported earlier (Kufel et al., RNA 1999) 18S rRNA processing is not affected by Rnt1 loss even though U14 snoRNA processing is affected.

9) Figure 6d. This is not the cleanest of data and I think we need to see an average of three repeats, with error bars, presented. Furthermore, I don't see the point of plotting uncleaved vs cleaved as in the 0 hour time-point there is nothing detectable for uncleaved. I think these should be plotted as percentage cleaved. I also feel that a Northern blot is required to show snR13 levels in the two strains for this experiment. Given the issues I mentioned above about GAL1::SNR13 expression (points 4 and 8 – see above) I would also like to see how the GAL1::SNR13-derived snoRNA compares at guiding rRNA methylation to the snR13 derived from its own promoter as the data presented suggests that this GAL1::SNR13-derived snoRNA is less efficient than the endogenous snR13 at guiding methylation. Birkedal et al., (Angew Chem Int Ed Engl, 2015) showed that snR13 methylated the rRNA to a high level – higher than that seen in the data presented in Figure 6. It is important to demonstrate that the GAL1::SNR13 generates a snoRNA that is as active as the snoRNA generated from the natural locus.

10) In the discussion, the authors make sweeping statements about the importance of correct 5' processing in snoRNP formation. However, these global statements are inaccurate as they only apply to the Rnt1-dependent snoRNAs. Indeed, the authors have missed a previously published paper that showed that correct 5' processing of the U24 box C/D snoRNA is not required for snoRNA function (Ooi et al., RNA 1998). This snoRNA is not Rnt1-dependent, as it is intron encoded, but this point does contradict the global statements made throughout the manuscript that the authors need to address. Therefore, the authors need to explain these points clearly in the text, in the model (Figure 7) and the overview figure.

11) Page 21, line 509. The authors state that "In general, cells require high levels of snoRNA to modify abundant ribosomal RNAs." This is not the case, in humans where most snoRNAs are between a few hundred to max. 10,000 copies per cell (Maxwell and Fournier, Ann Rev Biochem, 1995). The levels of snoRNAs are lower in yeast. However, the cellular levels of most small nuclear RNAs are significantly lower in yeast compared to human cells. Furthermore, on page 21, line 517, the authors state that "transposed into an intronic location, snoRNA transcribed without an m7G cap may be more efficiently expressed". However, in human the two most abundant snoRNAs (U3 – 500,000 copies per cell and U8 – 20,000 copies per cell) are transcribed from independent genes

with an m7G cap on the nascent transcript. Furthermore, if expression from an intron would generate more snoRNA a difference in expression levels should be observed in yeast where both independently transcribed and intron-encoded snoRNAs are present. This whole argument in the manuscript makes no sense as it stands and should be removed.

Minor points:

1) Figure S1A. In the legends it states that the secondary structures are shown on the left whereas they are on the right in the figure.

2) Figure 1C and D and Figure 3. I feel that this figure would benefit from having all of the panels running 5' to 3' (i.e. left to right). As it is some (e.g. snR64) are 5' to 3' (left to right) relative to the gene while others (e.g. snR47) run right to left. This point applies to other figures in the paper such as supp fig1 and Figure 3.

3) Figure 4. The legend is insufficient for me to follow the figure. I assume the RH in panels b and c refer to RNase H but I am left to guess this. I also do not understand how homogenous 5' ends were generated using RNase H and what oligos were used from what is written in the text (page 12, line 280)?

4) Page 3, line 51. "Display specific secondary structures associated with class-specific proteins". This makes no sense and should be re-written.

5) The authors use a lot of acronyms/abbreviations which are, to my knowledge, not that commonly used. One example is RDN (RNA degradation in the nucleus). There are so many of these used I found this confusing and unhelpful and I would hope that the authors would reduce the use of these acronyms in a revised manuscript if they hope it to be understandable to a broader audience.

6) Page 11, line 266. Figure 3 should be referred to as well here. Also, this whole paragraph belongs to the previous section in the text and figure 4a should really be part of Figure 3 as this work has nothing to do with cap retention – the theme of this section of the manuscript.

7) Page 11, line 273. When the gal promoter is transcribing a ncRNA can the sequence ahead of the ncRNA really be referred to as the UTR (untranslated region)? I accept that in the natural context this region would be part of the UTR of the protein coding gene but the use of UTR in this context is confusing.

Response to reviewer's comments (in blue):

Reviewer #1 (Remarks to the Author):

Previous analyses reported that most mature, independently transcribed, C/D snoRNAs tested lack TGM caps and to show 5' maturation that is variably dependent on cotranscriptional cleavage by the Rnt1 endonuclease or 5' digestion by the Xrn1 and Rat1 exonucleases. This was also reported to be the case for a smaller subset of H/ACA snoRNAs. The authors present evidence for 5' cleavage of a small number of additional C/D and H/ACA snoRNAs, based on 5' extensions in *rnt1Δ* strains. Circular PCR indicated that 5' extended pre-snoRNAs are also 3' immature. The authors conclude that this demonstrates that 5' processing precedes, and may be required for, 3' processing of C/D snoRNAs. This is supported by cis-acting mutation in SNR68 and catalytic mutation in Rnt1. The lack of capping activity reduced the impairment of 3' processing in a GAL-SNR13 fusion construct suggesting that presence of the cap might delay 3' processing. Further data indicate that capped pre-snoRNAs can partially leak to the cytoplasm and appear to be defective in Nop1 binding. Consistent with these findings, rRNA methyltransferase activity was also reduced.

Overall, there is a tendency to draw strong general conclusions from the behavior of individual, artificial constructs. However, the work makes a useful contribution to understanding snoRNA maturation and will be of interest and significance to the field. There are some points that should be addressed.

Specific comments

1) The CHIP data in Figure 1 are presented in the context of discussion of the role of Rnt1 in 5' processing, but would appear to be more consistent with its 3' function. There does not seem to be an obvious difference between snoRNAs predicted to show or lack 5' cleavage. Are there any systematic differences between the Rnt1 or CBC distribution on snoRNAs with predicted 5' Rnt1 cleavage sites and snoRNAs or snRNAs that lack 5' cleavage?

Response: Our analyses indicate that maturation of box C/D snoRNA and box H/ACA should be considered separately. However, since there are only 4 non-intronic, Rnt1-independent box C/D snoRNA, any systematic analyses on such a small group would be unreliable. Box H/ACA snoRNA display a similar imbalance: 6 are Rnt1-dependent and 21 are Rnt1-independent. Investigation of Rnt1 and CBC distribution for individual snoRNA did not reveal any significant differences between Rnt1-dependent and Rnt1-independent snoRNA.

2) Pre-snoRNAs stabilized by loss of Rnt1 were previously reported to carry TMG caps (Lee et al 2003). In this work, the pre-snoRNAs are reported to carry m7G caps in the Abstract but this does not appear to have been addressed experimentally. This is important because TMG caps are not expected to be bound by CBC. It would therefore be expected that CBC would be displaced from the pre-snoRNA prior to or at cotranscriptional hypermethylation. This would be very relevant to the CHIP data for CBC binding. Reduced CBC signal is interpreted as reflecting cleavage, but could also reflect hypermethylation. This point should be addressed.

Response: Trimethyl guanosine synthase (Tgs1) which is responsible for snoRNA cap hypermethylation is localised in the nucleolus. To our knowledge, there is no evidence that Tgs1 is recruited to transcribed snoRNA genes. Thus, TMG formation probably occurs post-transcriptionally and therefore should not have any impact on co-transcriptional recruitment of the Cap Binding Complex. Moreover, we did not observe any exacerbation of the 3' end processing defect in the *rnt1Δ tgs1Δ* double mutant which argues for post-transcriptional modification. We made adjustment in the text to address this point.

3) Previous analyses indicated that the level of 5' Rnt 1 cleavage was highly variable between snoRNA species. This should be discussed with respect to the data presented here.

Response: Indeed, there are differences in Rnt 1-sensitivity/dependence between snoRNA species. In our analyses, we consistently used snoRNA whose 5' end processing is significantly affected by Rnt 1 activity (e.g. Supplementary Fig. 3b).

4) P8: The authors refer to "3'extended, Rrp6-dependent pre-snr13 precursors". By this do the authors intend: "3'extended, pre-snr13 that accumulates in the absence of Rrp6"?

The, presumably corresponding, snR13 Me band in the northern is described in the figure legend as "precursors as with short oligoadenylated 3'extensions". How was this shown?

Response: Maturation pathways for pre-snr13 precursors (both native and expressed from *GAL1* promoter) have been described in details in Grzechnik and Kufel, *Mol. Cell* 2008.

We clarify this in the text and added the reference.

5) P9: "Overall, we conclude that the synthesis of box H/ACA snoRNA requires m⁷G cap. In contrast this structure is dispensable for box C/D snoRNA expression...." This seems to be a remarkably sweeping statement based on two artificial constructs.

Response: In the revised manuscript we extended this analysis to further examples. Accumulation of box H/ACA snR43 and snR46 was affected in the *ceg1-63* mutant which confirms previous observations for snR3 (Fig. 2e and Supplementary Fig. 2d). We also tested if m⁷G cap is required for two additional box C/D snoRNA – snR65 and snR68. However, removal of their 5' extensions strongly inhibited snR68 synthesis already in the WT strain. Only maturation of snR65 was informative and consistent with data obtained for snR13 (Fig. 2e and Supplementary Fig. 2d). However, since we did not test all yeast snoRNA, we rephrased and toned down potential overstatements in the manuscript.

6) Figure 6e: It would be important to show the relative expression of snR13 from the GAL1 and GAL1U constructs.

Response: The levels of snR13 and snR13e are similar up to 8 hours of induction (Supplementary Fig. 5f) where we observed the strongest differences in rRNA methylation. After 10-12 hours on galactose-containing medium, snR13 accumulation

was higher, reinforcing our observations that removal of the capped 5' extension is required for efficient snoRNA synthesis.

7) Since the 5' extended pre-snoRNAs are also 3' extended, is it clear that the 5' extensions rather than 3' extensions are responsible for the observed defects?

Response: To address this question, we analysed 25S rRNA methylation in the *rrp6Δ* strain. Deletion of the *RRP6* gene results in accumulation of 3' extended snoRNA while 5' ends are fully processed. Our RTLN-qP analysis did not detect any changes in methylation of 25S rRNA in *rrp6Δ* when compared to the WT (Supplementary Fig. 5a). Thus, we concluded that 3' extensions do not interfere with snoRNA functions in rRNA methylation.

Reviewer #2 (Remarks to the Author):

This is a nice study. The authors utilize a broad set of complementary tools to interrogate this aspect of snoRNA function, and the largely systematic and well-conceived experiments support the model presented in Figure 7. While I find the work significantly impressive, this is not my field and I am thus not qualified to comment on the potential impact of this work. Potential issues that might be addressed to improve the quality of the manuscript are listed below:

(1) While some of it had to do with my lack of expertise with this field, the difficulty that I had in reading this manuscript was also due to less-than-ideal editing and organization. It seems that approaches could be rationalized a bit more cleanly and some degree of discussion surrounding each of the drawn conclusions in the results section would be better presented in the discussion proper. While this latter point may only be my personal preference, I found the speculation around the conclusions disruptive to the otherwise strong logic flow in the work. Examples include but are not limited to lines 203-207 and 238-241 in the manuscript.

Response: The manuscript has been significantly re-edited. We have also toned down unnecessary speculation and improved the paper's logic flow.

(2) For figure 1, the logical argument presented in support of the RNA-independence of Rnt1 recruitment (lines 134-139) is reasonable, but direct demonstration of this would certainly strengthen the position. Perhaps this has already been directly demonstrated elsewhere?

Response: To address this comment, we immunoprecipitated chromatin associated with Rnt1 from RNase-treated extracts (Fig. 1f). Our new results show that Rnt1 co-transcriptional recruitment depends on RNA. However, since Rnt1 is present at the 3' ends of both Rnt1-dependent and independent snoRNA we concluded that Rnt1 recruitment is autonomous of RNA stem-loops in the 5' extensions. Thus, we speculate that Rnt1 may be recruited through NNS complex which is bound to the nascent RNA over transcriptional terminators.

(3) Also regarding the data within Figure 1 - While the conclusions drawn here (lines 162- 165) are generally supported by the data, I didn't quite follow the RNA stem-loop independence assertion. Perhaps this could be better described or phrased for the non-expert?

Response: As mentioned above, we have added new data and thoroughly re-written the manuscript. We believe that our revised manuscript is now more accessible to non-experts.

(4) Figure 2 - The methodological description of the *ceg1-63* induction experiments here is a bit unclear. Were the WT controls also assayed at the restrictive temperature? Is processing any different in the *ceg1-63* line at the permissive temperature? I am sure that the possibility that these observations are an artefact of temperature or strain were ruled out, but it would be an improvement if this were explicitly explained for the audience.

Response: We clarified in the text growth conditions used for this experiment. Moreover we performed *GAL1::SNR3* and *GAL1::SNR13* transcription induction in WT and the *ceg1-63* mutant at permissive temperature (25°C) (Supplementary Fig. 2C). We did not observe any significant differences in snR13 accumulation between WT and *ceg1-63* strains under these conditions. The levels of snR3 were slightly lower in *ceg1-63* when compared to the WT. However, snR3 synthesis was not impaired as observed at non-permissive temperature. Note that *ceg1-63* has a point mutation which may result in a mild phenotype at permissive temperature.

(5) Also regarding the data within Figure 2: The conclusion drawn from the data within this Figure (lines 203- 207) is that synthesis of box H/ACA snoRNAs requires a cap, and while structure is dispensable for box C/D snoRNA expression, where absence of the cap accelerates 3' end processing. Again, perhaps demonstrating my ignorance of this field, I am not sure that this conclusion can be broadly generalized as it appears to be here, particularly given that mutants not dependent on Rnt 1 were specifically chosen to be employed. This concern can actually be extended through much of the manuscript – several different box C/D and box H/ACA snoRNAs to utilized to make specific points. Perhaps like here (where I actually understood the rationale) this is very careful selection to best illustrate selected points due to specific snoRNA sequence or structure (I didn't find rationalizations elsewhere). Perhaps it was a function of what on hand. This strategy supports the assertion that several of the snoRNAs conform to the ultimate model, but that distinct snoRNAs are utilized to test the various aspects of the mechanism challenges the generalizability of the model at the same time. This is a potential weakness of the work – a strategy of consistently sticking with two, say snR68 and snR65, unless absolutely necessary throughout the manuscript would have been preferable.

Response: Our goal was to show that the process we describe is general, not specific for a particular snoRNA. Therefore, we used a broad selection of snoRNA examples. However, to support our observations and address the issue raised by the reviewer, we performed experiments where we induced transcription of Rnt 1-dependent *SNR65*, *SNR68*, *SNR43* and *SNR46* in the *ceg1-63* strain. These snoRNA are normally transcribed with a 5' extension, which is removed by Rnt 1 in the WT. To prevent possible co-

transcriptional degradation from the unprotected 5' end (see Fig. 4e) only snoRNA mature sequences were expressed from the integrated *GAL1* promoter.

Consistent with our previous results, synthesis of box H/ACA snR43 and snR46 was severely affected in the *ceg1-63* mutant (Fig. 2e and Supplemental Fig. 2d). Intriguingly, we did not obtain a high accumulation of box C/D snR65 transcript from the *GAL1* promoter either in WT or *ceg1-63* mutant. However, the observed level of snR65 in *ceg1-63* was unaffected when compared to WT (Fig. 2e). This is consistent with our data obtained for *SNR13* processing in *ceg1-63* and further indicates that m⁷G cap is not required for box C/D snoRNA synthesis.

We also attempted to test for maturation of snR68 transcribed from the *GAL1* promoter in WT and *ceg1-63*. However, after induction, we only detected the accumulation of long, most likely 3' extended and polyadenylated RNA species (Supplemental Fig. 2d). This underlines the importance of the 5' end extension. This region seems to be somehow required for Rnt1-dependent snoRNA synthesis. For example, 5' extensions may play a role in NNS recruitment, since CLIP data reveal Nrd1 binding to the 5' regions of some snRNA precursors [1].

(6) Conclusion for data described in Figure 3: "Overall these data show a clear interplay. . . and exosome-dependent processing for box C/D snoRNA (lines 255-257)." Any role for the exosome may be reasonably assumed, but was not explicitly demonstrated here. These set of data and the overall remainder of the conclusion are OK.

Response: Roles for nuclear exosome in maturation of independently transcribed snoRNA have been reported previously [2]. As described, the accumulation of specific precursors may be directly associated with particular processing enzymes.

(7) Figure 4: This was really systematic, good job. What the *GAL1* 5' UTR experiments were explicitly testing and what the conclusion drawn from these data could be clarified and better integrated to the overall narrative. I'm not sure that I entirely followed that aspect of the logic here.

Response: We tried to improve the logic flow of the manuscript here.

(8) Figure 5: The authors conclude that cap removal is critical for nuclear localization and snRNP assembly. This summary can be deduced from the previous data, but that's not what was directly tested here. If this specific conclusion is to be drawn, the finding would be strengthened by repetition of key experiments in *ceg1-63* and *cbp80Δ* mutants.

Response: We did not observe any impact of *CBP20*, *CBP80* or double *CBP20/80* deletion on snoRNA processing (Fig. 2f, 4g and data not shown) therefore, testing snoRNA localization in *cbp80Δ* mutant would not contribute to our model. We concluded that m⁷G cap, not the Cap Binding Complex affects snoRNA processing. Localization of 5' unprocessed snoRNA in *ceg1-63* mutant would strongly support our data. However, such unprocessed snoRNA accumulates and mislocalizes only in *rnt1Δ* strain. In the strain where Rnt1 is present, the signal from 5' unprocessed snoRNA is localised in the nucleus and most likely originates from precursors still associated with the transcription sites (Fig. 5a).

Unfortunately, our attempts to introduce a *RNT1* deletion in *ceg1-63* were unsuccessful. As an alternative approach, we tested the localization of 5' extended snR13e in *ceg1-63* after transcriptional induction at non-permissive conditions. However, since the levels of snR13e in *ceg1-63* are very low due to 5'-3' degradation (Fig.4e), we were not able to obtain any informative results.

We have changed the text to clarify that there is a possibility that it is not m⁷G cap itself but rather an unknown factor(s) interacting with m⁷G that regulates snoRNA synthesis.

(9) It seems that the last section of the discussion, which focuses on snoRNA genomic organization as a function of maturation mechanism during evolution, would I be strengthened by an explicit comparison of box C/D and box H/ACA organization from cerevisiae to sapiens. This seems to be the test of the idea put forth – wouldn't one expect that more box C/D than box H/ACA have wound up in introns due to the selective pressure exerted by the mechanisms described here on the former? If this isn't true, I'm not sure that I would make the argument at all - the H/ACA caveat at the end doesn't seem to salvage it. Beyond this, the model here was cerevisiae, and so perhaps the mechanism within human cells should be evaluated prior to drawing these parallels.

Response: A single snapshot of human snoRNA gene organization does not reveal any preference towards intronic localization for either snoRNA class. Such a bias may be revealed by a comprehensive phylogenetic analysis employing multiple organisms appearing in a different point of the evolutionary tree. However, such an in-depth analysis is beyond the scope of our current study.

We toned down this conclusion to make it clear that this is purely speculation.

Reviewer #3 (Remarks to the Author):

In this manuscript, Grzechnik and co-authors, have taken a global approach to study the 5' processing of snoRNAs and its importance to snoRNA accumulation and function. The authors conclude that 5' processing is important for snoRNA processing and function and propose a mechanism by which Rnt1 cleavage 5' of the snoRNA, which removes the m⁷G cap structure which is added co-transcriptionally, is needed for efficient snoRNA processing, association with snoRNP proteins, correct localisation and snoRNP function. This is, in principle, an interesting story. However, this manuscript comes across as a rushed first draft and I feel that the way this manuscript has been prepared has made it difficult to judge the data. A significant part of the manuscript focuses on Rnt1 cleavage of pre-snoRNAs yet little is done to explain which elements of this paper are new and which confirm earlier work on Rnt1 and snoRNA processing. Furthermore, the authors make sweeping statements about box C/D snoRNA processing and function based on their data when, in fact, the phenomenon they describe are only seen with a subset of snoRNAs, even if they are the majority.

Major Points:

1) The authors do not clearly explain in the introduction the different snoRNA gene types and which are found in humans (primarily intronic), plants (intronic and polycistronic), and yeast (mainly mono-cistronic with some intronic and some polycistronic).

Furthermore, authors make global sweeping statements about yeast snoRNAs, such as the claim that mature yeast box C/D snoRNAs lack a cap (page 3, line 69). This is clearly not the case as 4 mature snoRNAs, U3, snR13, snR4 and snR45 (Balakin et al., Cell 1996) have 5' cap structures. Indeed, the authors manage to contradict themselves by stating this themselves in the results section. Furthermore, one yeast snoRNA, snR52 (which the authors analyse in the manuscript), is transcribed by RNA pol III, a point missed by the authors. The introduction therefore needs to be re-written so that these points are clear to the readers.

Response: We have adjusted the introduction to better describe the diversity of genomic snoRNA organization. We also named Rnt 1-dependent snoRNA (RD snoRNA) to exclude 4 exceptions from our model.

We had mentioned that snR52 is synthesised by Pol III and therefore acts as a control for normally Pol II transcribed snoRNA. We have emphasised this point throughout the manuscript so that it will be evident to the reader.

2) Throughout the manuscript, the authors make sweeping statements about box C/D snoRNPs that are not true and need explaining properly. The observation that correct 5' processing by Rnt 1 is required for snoRNP formation and function only applies to a subset of the box C/D snoRNAs, even if it is the major group, and the text needs to be adjusted to reflect this.

Response: In the revised version of the manuscript we re-named the major group of box C/D snoRNA as Rnt 1-dependent snoRNA (RD snoRNA) to exclude the 4 snoRNAs that are not 5' end processed by Rnt 1.

3) Page 5, line 112. The authors claim that 3 snoRNAs retain their cap structures. However, 4 snoRNAs, U3, snR13, snR4 and snR45, have been shown to be capped with an m³G cap (Balakin et al., Cell 1996). Why is U3 omitted from table 1 and not discussed at all in the manuscript? U3 is assembled and processed by the same machinery, as far as we are aware, as the other snoRNAs so this omission makes no sense at all.

Response: We excluded snR17 (U3) from our analysis since it undergoes a specific maturation pathway. Notably it contains an intron and is processed by the splicing machinery. Moreover snR17a/b is processed at its 3' end by an snRNA-specific pathway mediated by Lhp1 proteins [3].

However, we agree that snR17 should be mentioned in the context of cap presence. We have therefore changed the text accordingly.

4) Figure 2A. The authors claim that Mt represents decapped, 5' truncated transcripts. However, these transcripts are present even before induction of gene expression (time-point 0) and, for the wild-type, do not vary upon induction of gene expression. The authors claim that this is due to basal activity of the gal promoter. If this is the case, why is only the mt form produced, this makes no sense. This observation must be explained by the authors and for me to believe that these are really snR13 transcripts, and not just some cross-reaction of the probe used. Furthermore, if these are snR13 transcripts, I would also need to see whether these transcripts occur at significant levels when snR13 is transcribed from its normal promoter and that this is not some artefact of the Gal1 system used in this experiment.

Response: The Mt snR13 transcripts were described in earlier studies from the Culbertson lab [4] and have also been observed in other studies [5-7].

The authors claim that there is only a slight decrease in mature snR13 levels in the *ceg1-63* strain. I would need to see quantitation of this experiment before I would believe this.

Response: Quantitation is presented in the Fig. 2d. Note that all RNA species throughout our study have been quantified.

Furthermore, since the authors claim that alternative 3' processing phenotype was seen in the *ceg1-63* strain, it would be good to know whether this process is dependent on Rrp6.

Response: In order to address this comment, we deleted *RRP6* in the *ceg1-63 GAL1::SNR13* strain and induced transcription of *SNR13* in non-permissive temperature (Supplementary Fig. 2a). We observed similar accumulation of Me and Pa precursors in *rrp6Δ* and *rrp6Δ ceg1-63* strains. However, some longer polyadenylated precursors or poly(A) tails were reduced in the *rrp6Δ ceg1-63* double mutant. This indicates that uncapped pre-snR13 are also processed by the core exosome.

In Figure S2A the authors used RNase H treatment to characterise the transcripts. Why in this experiment is the Mt form missing?

Response: We improved the quality of this Northern Blot (Supplementary Fig. 2b)

Finally, the authors conclude (page 9, line 203) that the cap structure is dispensable for box C/D snoRNA expression. However, all the data presented for the observation in the paper and the conclusions drawn are based on one snoRNA from each class of snoRNA. It is also possible that the snoRNAs that undergo 5' Rnt1 processing are dependent on the m⁷G cap – a point that should be tested or, if the authors have data on this, discussed in the manuscript.

Response: As with reviewer 1, to support our observations and address this issue raised, we performed experiments where we induced transcription of Rnt1-dependent *SNR65*, *SNR68*, *SNR43* and *SNR46* in *ceg1-63* strain. These snoRNA are normally transcribed with 5' extension, removed by Rnt1 in the WT. To prevent possible co-transcriptional degradation from such unprotected 5' ends (see Fig. 4e) only snoRNA mature sequences were expressed from the *GAL1* promoter.

Consistent with our previous results, synthesis of box H/ACA snR43 and snR46 was severely affected in *ceg1-63* mutant (Fig. 2e and Supplemental Fig.2d). Significantly, we could not obtain high accumulation of box C/D snR65 transcribed from *GAL1* promoter either in WT or *ceg1-63* mutant. However, the observed level of snR65 in *ceg1-63* was unaffected as compared to WT (Fig.2e). This is consistent with our data obtained for *SNR13* processing in *ceg1-63* and further indicates that m⁷G cap is not required for box C/D snoRNA synthesis.

We also attempted to test snR68 transcripts from the *GAL1* promoter in WT and *ceg1-63*. However, after induction, we observed only accumulation of long, most likely 3' extended and polyadenylated species (Supplemental Fig.2d). This additionally underlies

the importance of late co-transcriptional Rnt1 cleavage as the 5' extension appears to be somehow required for snoRNA synthesis. For example, these extensions may play a role in NNS recruitment since CLIP data reveal Nrd1 binding to the 5' regions of some snRNA precursors [1].

5) Figure 3. The authors explain that "box C/D snoRNA transcribed with 5' extensions as well as the last snoRNA from polycistronic TUs have short unprocessed extensions at their 3' ends". The data needed to back up this statement is not presented. Indeed, data is presented for snR51 (I assume, the labels are insufficient to determine whether this is snR51 or the other members of the cluster) but not the rest of the cluster. The authors need to present more data to validate this statement.

Response: We added more examples (Supplementary Fig. 3a) showing snoRNA 3' ends in *rnt1Δ*. We also added diagrams indicating the location of the shown regions.

6) Page 13, line 320. "Overall, our analysis indicate that the presence of the m7G cap but not the associated CBC on the box C/D snoRNA precursor interferes with the final step of 3' processing." While this is fine the authors do not really propose why this is not a problem with the 4 box C/D snoRNAs that retain the 5' cap structure. The same statement is also made on page 15, line 361. The authors must clarify that this only applies to a subset of the snoRNAs, even if it is the majority, and not all box C/D snoRNAs.

Response: As mentioned above we have re-named the major group of box C/D snoRNA as Rnt1-dependent snoRNA (RD snoRNA) to exclude the 4 snoRNA that are not 5' end processed by Rnt1 cleavage.

7) Figure 5. Mature snR68 should be in the nucleolus, a cap-like structure within the nucleus, that does not stain well with DAPI. However, the mature snR68 is found throughout the nucleus in Figure 5A. Indeed, the same is seen for snR43 and snR13 (panel c). In contrast, snR13 in panel f, and Nop1 (panel d), a protein associated with both snR13 and snR68, are found in nucleoli and are present a cap at the edge of the DAPI stained region. Unfortunately, in addition I cannot see the differences the authors describe in the text in panel f. The *GAL1::SNR13* and the *GAL1U::SNR3::SNR13* snoRNAs do not localise the same as the WT snR13, even though they should for the experiment to be valid. I am therefore not satisfied with the quality of the FISH data and find it difficult to draw any conclusions from the experiments.

Response: snoRNA trafficking in the nucleus has not been studied thoroughly and only for limited number of snoRNA, usually for U3, rarely snR10 snR30 or U14 [8-11]. Therefore, we cannot predict if analysed Rnt1-dependent snoRNA localise exclusively to the nucleolus, especially taking into account their complex maturation pathways. We consider that we have observed the physiological distribution of these snoRNA in *S. cerevisiae* and have changed the text to emphasise this. Moreover, to improve visualization of the FISH analysis we marked cell walls with dotted lines. This clearly shows that the fluorescent signal in the *rnt1Δ* strain is spread throughout the cell nucleus and cytoplasm.

SnR13 is overexpressed from the *GAL1* promoter and therefore its nuclear distribution may be perturbed. However, the rationale behind this experiment was to show differences between snR13, snR13e and snR13-snR3 expressed from *GAL1* promoter.

In the IP experiment (panel e), the authors claim that Nop1 was only associated with snoRNAs that were shortened at their 5' ends. However, while the pre-snoRNAs that were shortened at their 5' ends were enriched in the Nop1 IP material, the longer forms were also present and the data is not as "black and white" as the authors describe in the text. Furthermore, I think that the binding of Nop58 to the snoRNAs/pre-snoRNAs needs to be tested as this protein is essential for snoRNA formation and binds directly to the snoRNA. In contrast, structural data suggest that Nop1 primarily associates with the snoRNAs through its interaction with Nop56 and Nop58.

Response: We have removed this small data set from our paper as we could not confirm the Nop1 and Nop56 RIP-seq data by Northern Blot analysis.

8) Figure 6a, b and c. The major issue I have with the work presented in this figure is that Rnt1 is important for rRNA processing. This point is not mentioned in the manuscript, which is surprising given the work performed earlier by Kufel on the subject. Lack of Rnt1 has an impact on ribosome biogenesis that could therefore non-specifically affect the results presented in this figure irrespective of Rnt1's role in snoRNA maturation. These points must be acknowledged in the text and controls are needed to verify the specificity of the observations presented. Indeed, in Kufel et al., (RNA, 1999) it was shown that 25S production was reduced in the absence of Rnt1. It might therefore make more sense to look at methylation in the 18S rRNA. The authors have indicated that there are three types of box C/D snoRNA; capped (e.g. snR13), those that are Rnt1 – dependent in 5' processing and those that are Rnt1 - independent (a decent table emphasising this point would be appreciated). Therefore, loss of Rnt1 should only affect those that require Rnt1 for the correct 5' end processing of the snoRNA. In order to demonstrate that only these snoRNAs are affected the authors must show methylation data for individual snoRNAs of each type in control cells and cells lacking Rnt1. This could be achieved through a high-throughput sequencing approach but I would be happy if the authors would use the RNase H approach, established by the Steitz lab, to analyse the activity a couple of examples of Rnt1-dependent and –independent snoRNAs. I would also like to see the impact of Rnt1 on U14 levels and methylation activity. U14 is required for both rRNA methylation and 18S rRNA processing. If U14 is significantly impacted by Rnt1 loss then both rRNA methylation and 18S rRNA processing should be affected. However, as reported earlier (Kufel et al., RNA 1999) 18S rRNA processing is not affected by Rnt1 loss even though U14 snoRNA processing is affected.

Response: To confirm our results, we analysed 4 individual methylation sites in both 25S and 18S rRNA in WT and *rnt1Δ* cells. However, instead of using the suggested RNase H-dependent approach (which is not established in our lab) we instead employed the DNazyme-dependent method. This is less sensitive than directed RNase H cleavage and therefore detects only significant changes in methylation [12]. Our analyses of snR56 and snR72-dependent methylation sites in 18S rRNA and snR68- and snR13-dependent sites in 25S rRNA (Fig. 6e) revealed that methylation catalysed by Rnt1-dependent snR56, snR72 (in 18S rRNA) and snR68 (in 25S rRNA) is decreased in *rnt1Δ* strain. In contrast, methylation levels catalysed by Rnt1-independent snR13 were unaffected by *RNT1* deletion.

DNAzymes cleave RNA between purine and pyrimidine or between an NG dinucleotide [12]. Therefore, the sequence methylated by snR128 (U14) is unsuitable for our assay. The RTLN-qP analysis was not applicable either since there are other methylation sites close to the U14-dependent nucleotide. Kufel et al., RNA 1999 reported that although the absence of Rnt1 does not block cleavages in the 5' external transcribed spacers, the early pre-rRNA cleavages are kinetically delayed [13]. This is consistent with our data as we do not observe a total shutdown of snoRNA functions. Note that rRNA is still methylated to some extent in *rnt1Δ* strain. This is result of alternative processing which removes the 5' extension in the absence of Rnt1 as mentioned several times in the manuscript. Moreover, we also observed high levels U14 snoRNA in the *rnt1Δ* strain as compared to other snoRNA (Supplementary Fig. 5c). Therefore, we predict that the amount of alternatively processed U14 may be sufficient to mediate U14-dependent methylation and cleavage of pre-rRNA. Alternatively, these processes are mediated by 5' processed/truncated snR190-128 dicistron. We have made changes in the text to address these reviewer comments.

9) Figure 6d. This is not the cleanest of data and I think we need to see an average of three repeats, with error bars, presented. Furthermore, I don't see the point of plotting uncleaved vs cleaved as in the 0 hour time-point there is nothing detectable for uncleaved. I think these should be plotted as percentage cleaved. I also feel that a Northern blot is required to show snR13 levels in the two strains for this experiment. Given the issues I mentioned above about GAL1::SNR13 expression (points 4 and 8 – see above) I would also like to see how the GAL1::SNR13-derived snoRNA compares at guiding rRNA methylation to the snR13 derived from its own promoter as the data presented suggests that this GAL1::SNR13-derived snoRNA is less efficient than the endogenous snR13 at guiding methylation. Birkedal et al., (Angew Chem Int Ed Engl, 2015) showed that snR13 methylated the rRNA to a high level – higher than that seen in the data presented in Figure 6. It is important to demonstrate that the GAL1::SNR13 generates a snoRNA that is as active as the snoRNA generated from the natural locus.

Response: We plotted % cleaved as suggested. We also show Northern Blots indicating the levels of snR13 and snR13e. We made changes to the manuscript addressing this comment of the reviewer.

We also analysed the snr13-dependent methylation site in WT vs *GAL1::SNR13* strain using the DNAzyme-dependent assay (Supplementary Fig. 5e). However, we did not detect any differences in snR13-dependent 25S rRNA methylation between the strains.

10) In the discussion, the authors make sweeping statements about the importance of correct 5' processing in snoRNP formation. However, these global statements are inaccurate as they only apply to the Rnt1-dependent snoRNAs. Indeed, the authors have missed a previously published paper that showed that correct 5' processing of the U24 box C/D snoRNA is not required for snoRNA function (Ooi et al., RNA 1998). This snoRNA is not Rnt1-dependent, as it is intron encoded, but this point does contradict the global statements made throughout the manuscript that the authors need to address. Therefore, the authors need to explain these points clearly in the text, in the model (Figure 7) and the overview figure.

Response: In this manuscript, we focused on independently transcribed snoRNA units. Since they are all transcribed by RNA Pol II, the same polymerase responsible for mRNA synthesis, they must employ a dedicated co-transcriptional mechanism to facilitate snoRNA-specific maturation (eg. NNS transcription termination complex). In contrast, intronic snoRNA lack independent TSS, TTS and m⁷G cap. To clarify this we underline that our observed effect are specific to independently transcribed snoRNA.

11) Page 21, line 509. The authors state that “In general, cells require high levels of snoRNA to modify abundant ribosomal RNAs.” This is not the case, in humans where most snoRNAs are between a few hundred to max. 10,000 copies per cell (Maxwell and Fournier, Ann Rev Biochem, 1995). The levels of snoRNAs are lower in yeast. However, the cellular levels of most small nuclear RNAs are significantly lower in yeast compared to human cells. Furthermore, on page 21, line 517, the authors state that “transposed into an intronic location, snoRNA transcribed without an m⁷G cap may be more efficiently expressed”. However, in human the two most abundant snoRNAs (U3 – 500,000 copies per cell and U8 – 20,000 copies per cell) are transcribed from independent genes with an m⁷G cap on the nascent transcript. Furthermore, if expression from an intron would generate more snoRNA a difference in expression levels should be observed in yeast where both independently transcribed and intron-encoded snoRNAs are present. This whole argument in the manuscript makes no sense as it stands and should be removed.

Response: Direct comparison of snoRNA absolute levels between such distinct organisms as human and yeast is questionable. Thus, the human genome is much more complex and gene expression varies between different cell types. Furthermore, the number of snoRNA genes in human cells is hard to establish. In yeast, there are two canonical classes while in human cells there is a great variety of snoRNA variants. While the example suggested by the reviewer is correct, we would point out that in the case of U3, this snoRNA is essential and therefore may require specific gene expression regulation which does not apply to other snoRNA. In general, in all biological systems there will be exceptions. For example, while most mRNA in mammals is 3' end polyadenylated, mRNA of replication-dependent histones is not, affording a more direct control of gene expression.

We do appreciate the reviewer's concerns and have consequently toned down this part of the manuscript.

Minor points:

1) Figure S1A. In the legends it states that the secondary structures are shown on the left whereas they are on the right in the figure.

Response: Corrected.

2) Figure 1C and D and Figure 3. I feel that this figure would benefit from having all of the panels running 5' to 3' (i.e. left to right). As it is some (e.g. snR64) are 5' to 3' (left to right) relative to the gene while others (e.g. snR47) run right to left. This point applies to other figures in the paper such as supp fig1 and Figure 3.

Response: Corrected.

3) Figure 4. The legend is insufficient for me to follow the figure. I assume the RH in panels b and c refer to RNase H but I am left to guess this. I also do not understand how homogenous 5' ends were generated using RNase H and what oligos were used from what is written in the text (page 12, line 280)?

Response: We have improved the figure.

4) Page 3, line 51. "Display specific secondary structures associated with class-specific proteins". This makes no sense and should be re-written.

Response: Corrected.

5) The authors use a lot of acronyms/abbreviations which are, to my knowledge, not that commonly used. One example is RDN (RNA degradation in the nucleus). There are so many of these used I found this confusing and unhelpful and I would hope that the authors would reduce the use of these acronyms in a revised manuscript if they hope it to be understandable to a broader audience.

Response: We reduced the amount abbreviations in the text, replacing them with full names.

6) Page 11, line 266. Figure 3 should be referred to as well here. Also, this whole paragraph belongs to the previous section in the text and figure 4a should really be part of Figure 3 as this work has nothing to do with cap retention – the theme of this section of the manuscript.

Response: We add the reference but decided to keep the formatting as is. In our opinion such sectioning is logical.

7) Page 11, line 273. When the gal promoter is transcribing a ncRNA can the sequence ahead of the ncRNA really be referred to as the UTR (untranslated region)? I accept that in the natural context this region would be part of the UTR of the protein coding gene but the use of UTR in this context is confusing.

Response: We agree with the Reviewer that a native 5' upstream fragment cannot be referred as an UTR in the context of ncRNA. However, in this experiment, the sequence 5' to the ncRNA is the *GAL1* gene UTR. This sequence was synthesized and introduced to the genome together with the *GAL1* promoter. We clarify in the text that the UTR 5' to the snoRNA is the *GAL1* UTR.

References:

1. Jamonnak, N., et al., *Yeast Nrd1, Nab3, and Sen1 transcriptome-wide binding maps suggest multiple roles in post-transcriptional RNA processing*. RNA, 2011. **17**(11): p. 2011-25.
2. Grzechnik, P. and J. Kufel, *Polyadenylation linked to transcription termination directs the processing of snoRNA precursors in yeast*. Mol Cell, 2008. **32**(2): p. 247-58.
3. Kufel, J., et al., *A complex pathway for 3' processing of the yeast U3 snoRNA*. Nucleic Acids Res, 2003. **31**(23): p. 6788-97.
4. Rasmussen, T.P. and M.R. Culbertson, *The putative nucleic acid helicase Sen1p is required for formation and stability of termini and for maximal rates of synthesis and levels of accumulation of small nucleolar RNAs in Saccharomyces cerevisiae*. Mol Cell Biol, 1998. **18**(12): p. 6885-96.
5. Ganem, C., et al., *Ssu72 is a phosphatase essential for transcription termination of snoRNAs and specific mRNAs in yeast*. EMBO J, 2003. **22**(7): p. 1588-98.
6. Dichtl, B., et al., *A role for SSU72 in balancing RNA polymerase II transcription elongation and termination*. Mol Cell, 2002. **10**(5): p. 1139-50.
7. Feigenbutz, M., et al., *The exosome cofactor Rrp47 is critical for the stability and normal expression of its associated exoribonuclease Rrp6 in Saccharomyces cerevisiae*. PLoS One, 2013. **8**(11): p. e80752.
8. Qu, L.H., et al., *Seven novel methylation guide small nucleolar RNAs are processed from a common polycistronic transcript by Rat 1p and RNase III in yeast*. Mol Cell Biol, 1999. **19**(2): p. 1144-58.
9. Verheggen, C., et al., *Mammalian and yeast U3 snoRNPs are matured in specific and related nuclear compartments*. EMBO J, 2002. **21**(11): p. 2736-45.
10. Verheggen, C., et al., *Box C/D small nucleolar RNA trafficking involves small nucleolar RNP proteins, nucleolar factors and a novel nuclear domain*. EMBO J, 2001. **20**(19): p. 5480-90.
11. Narayanan, A., et al., *Role of the box C/D motif in localization of small nucleolar RNAs to coiled bodies and nucleoli*. Mol Biol Cell, 1999. **10**(7): p. 2131-47.
12. Buchhaupt, M., C. Peifer, and K.D. Entian, *Analysis of 2'-O-methylated nucleosides and pseudouridines in ribosomal RNAs using DNazymes*. Anal Biochem, 2007. **361**(1): p. 102-8.
13. Kufel, J., B. Dichtl, and D. Tollervey, *Yeast Rnt 1p is required for cleavage of the pre-ribosomal RNA in the 3' ETS but not the 5' ETS*. RNA, 1999. **5**(7): p. 909-17.

Reviewers' Comments:

Reviewer #1:

Remarks to the Author:

The authors have adequately revised and improved the MS and I am willing to recommend publication. The snoRNAs form quite a topical subject, so this report on the significance of their processing should be of quite wide interest.

Reviewer #2:

Remarks to the Author:

With the exception of the retention of a few distracting digressions (lines 148-152 and 264-270, as well as the evolutionary argument at the end), the manuscript is much improved and has adequately addressed the issues that I raised for the initial submission. In regards to the evolutionary discussion, the authors rebutted that a full analysis would be beyond the scope of the work. I agree with this, but then would also ask the authors whether then that the speculation, softened as it is, should be retained in this case.

Reviewer #3:

Remarks to the Author:

In general, the authors have satisfactorily addressed the majority of my points. However, the following points have not been addressed:

Major point 1. The authors now claim in their response that snR52 is mentioned throughout the manuscript. However, the first time I see snR52 described as a pol III transcription is on page 13 in the results. Please can the authors mention in the introduction that this is the one RNA pol III transcribed snoRNA. It is strange that the authors mention that there is a RNA pol III transcribed snoRNA in the introduction but without telling us what snoRNA this is.

Major point 7. While I agree that the localization of only a few yeast snoRNAs have been characterized I feel that stating that the some of the snoRNAs localize predominantly to the nucleoplasm naturally needs more controls than provided in this manuscript. For example, have the authors tested the FISH probes on cells lacking the snoRNAs analyzed? I agree that this was not the point of the experiment but if there are questions about the localization of the WT snoRNA I do not understand how any conclusions can be drawn. The Rnt1-dependent snoRNAs represent the majority of the box C/D snoRNAs and they need core proteins, such as Nop58, for their stability. Furthermore, based on earlier work, I would expect the majority of the snoRNAs in question to be associated with Nop1. Both Nop1 and Nop58 are present almost exclusively in the nucleolus. Therefore, I would expect the majority of each box C/D snoRNA to also localize to the nucleolus. While the authors could be correct that these snoRNAs naturally localize outside the nucleolus, I feel that their data does not fit with the published literature on this point and therefore more controls are needed before this data is suitable for publication.

Minor point 5. While the authors have tried to reduce the use of abbreviations I still feel that there are too many specialized abbreviations (e.g. NNS) used in the manuscript. While reading the revised version I had to constantly keep referring to the introduction to work out what the abbreviations meant which made reading the paper difficult and will make this work very difficult for a non-expert to read. Please reduce further the use of abbreviations.

Reviewer #3 (Remarks to the Author): *Author responses in italics*

In general, the authors have satisfactorily addressed the majority of my points. However, the following points have not been addressed:

Major point 1. The authors now claim in their response that snR52 is mentioned throughout the manuscript. However, the first time I see snR52 described as a pol III transcription is on page 13 in the results. Please can the authors mention in the introduction that this is the one RNA pol III transcribed snoRNA. It is strange that the authors mention that there is a RNA pol III transcribed snoRNA in the introduction but without telling us what snoRNA this is.

Response: In the final revised version of the manuscript, we have now added reference to the SNR52 Pol III transcribed gene in the introduction (line 61).

Major point 7. While I agree that the localization of only a few yeast snoRNAs have been characterized I feel that stating that the some of the snoRNAs localize predominantly to the nucleoplasm naturally needs more controls than provided in this manuscript. For example, have the authors tested the FISH probes on cells lacking the snoRNAs analyzed? I agree that this was not the point of the experiment but if there are questions about the localization of the WT snoRNA I do not understand how any conclusions can be drawn. The Rnt1-dependent snoRNAs represent the majority of the box C/D snoRNAs and they need core proteins, such as Nop58, for their stability. Furthermore, based on earlier work, I would expect the majority of the snoRNAs in question to be associated with Nop1. Both Nop1 and Nop58 are present almost exclusively in the nucleolus. Therefore, I would expect the majority of each box C/D snoRNA to also localize to the nucleolus. While the authors could be correct that these snoRNAs naturally localize outside the nucleolus, I feel that their data does not fit with the published literature on this point and therefore more controls are needed before this data is suitable for publication.

Response: The majority of snoRNA localization analyses previously described have been on U3 snoRNA, which clearly does localize to the nucleolus. Similarly, as shown in our IF analysis (Figure 5), wild-type snR13 is also present in the nucleolus (e.g. Figure 5E). However, our IF analysis is too low resolution to accurately distinguish nucleolar versus nuclear localisation for snoRNA processed at their 5' ends. Even so in our IF analysis we observe a wider nuclear localization of tested snoRNA implying more general nuclear presence. The focus of our analysis is to distinguish nuclear and cytoplasmic fractions. Our goal is to compare localization of 5' processed and 5' unprocessed snoRNA showing that lack of Rnt1 processing causes cytoplasmic accumulation. To validate our data, we have now included the suggested additional FISH analysis with probes targeting snR68 in the SNR68 deletion strain (snR68Δ). This experiment indicates that anti-snR68 probes did not have off-target specificity (see new Supplementary Figure 5a and associated text).

We agree that there may be a discrepancy between Rnt1-dependent box C/D snoRNA and Nop1/Nop58 localization. Nop1 and Nop58 exclusively nucleolar localization may be determined by nucleolar localization of very abundant U3 and other snoRNA (e.g. snR13) which are not 5' end processed. Since U3 levels are much higher than for other snoRNAs^{1,2} and accumulate in a very small and defined area in the nucleolus, it is possible that the previously observed Nop1 and Nop58 nucleolar specific signals is U3 associated and therefore may overshadow dispersed wider nuclear signal.

Biochemical analysis published elsewhere clearly indicate that various snoRNA display distinct functional properties which may mirror differential nuclear localization. For example, ribosome fractionation and analyses of associated snoRNA show that U3 snoRNA is strongly associated with pre-rRNA while other snoRNA (e.g. snR47, snR79 or snR57) are mainly present in the cell as a free pool (Reviewer response Figure below)³⁻⁵. Although this is not direct evidence for subcellular localization, these data do indicate that snoRNA are differentially linked to rRNA synthesis and therefore may be present in various nuclear compartments.

[Redacted]

Minor point 5. While the authors have tried to reduce the use of abbreviations I still feel that there are too many specialized abbreviations (e.g. NNS) used in the manuscript. While reading the revised version I had to constantly keep referring to the introduction to work out what the abbreviations meant which made reading the paper difficult and will make this work very difficult for a non-expert to read. Please reduce further the use of abbreviations.

Response: We have attempted to reduce abbreviation usage to a minimum. Names for complexes including CBC or NNS are however widely used and recognised in the RNA field. For example, NNS is widely referred to as the Nrd1-Nab3-Sen1 complex e.g. in recent papers published in *Molecular Cell*, *Genes and Developments*, *PLoS Genetics* and *Nature Reviews in Molecular Cell Biology*⁶⁻¹⁰. We feel that further reduction of abbreviations for example by replacing NNS with its full name Nrd1-Nab3-Sen1 is not appropriate.

References:

1. Tyc, K. & Steitz, J.A. U3, U8 and U13 comprise a new class of mammalian snRNPs localized in the cell nucleolus. *EMBO J* **8**, 3113-9 (1989).
2. Samarsky, D.A. & Fournier, M.J. Functional mapping of the U3 small nucleolar RNA from the yeast *Saccharomyces cerevisiae*. *Mol Cell Biol* **18**, 3431-44 (1998).
3. Kos, M. & Tollervey, D. The Putative RNA Helicase Dbp4p Is Required for Release of the U14 snoRNA from Preribosomes in *Saccharomyces cerevisiae*. *Mol Cell* **20**, 53-64 (2005).
4. Liang, X.H., Liu, Q. & Fournier, M.J. rRNA modifications in an intersubunit bridge of the ribosome strongly affect both ribosome biogenesis and activity. *Mol Cell* **28**, 965-77 (2007).
5. Sharma, S. et al. Specialized box C/D snoRNPs act as antisense guides to target RNA base acetylation. *PLoS Genet* **13**, e1006804 (2017).
6. Tudek, A. et al. Molecular basis for coordinating transcription termination with noncoding RNA degradation. *Mol Cell* **55**, 467-81 (2014).
7. Porrua, O. & Libri, D. Transcription termination and the control of the transcriptome: why, where and how to stop. *Nat Rev Mol Cell Biol* **16**, 190-202 (2015).
8. Lemay, J.F. et al. The Nrd1-like protein Seb1 coordinates cotranscriptional 3' end processing and polyadenylation site selection. *Genes Dev* **30**, 1558-72 (2016).
9. Fasken, M.B., Laribee, R.N. & Corbett, A.H. Nab3 facilitates the function of the TRAMP complex in RNA processing via recruitment of Rrp6 independent of Nrd1. *PLoS Genet* **11**, e1005044 (2015).
10. Chen, X. et al. Transcriptomes of six mutants in the Sen1 pathway reveal combinatorial control of transcription termination across the *Saccharomyces cerevisiae* genome. *PLoS Genet* **13**, e1006863 (2017).